# Formation of spatial vegetation patterns in heterogeneous environments

**Karl Kästner**[1]*, **Daniel Caviedes-Voullième**[2,3], **Christoph Hinz**[1]

**1** Hydrology, Brandenburg University of Technology Cottbus–Senftenberg, Cottbus, Germany, **2** Institute of Bio- and Geosciences: Agrosphere (IGB-3), Forschunszentrum Jülich, Jülich, Germany, **3** Simulation and Data Lab Terrestrial Systems, Jülich Supercomputing Centre (JSC), Jülich, Germany

* karl.kaestner@b-tu.de

## Abstract

Functioning of many resource-limited ecosystems is facilitated through spatial patterns. Patterns can indicate ecosystems productivity and resilience, but the interpretation of a pattern requires good understanding of its structure and underlying biophysical processes. Regular patterns are understood to form autogenously through self-organization, for which exogenous heterogeneities are negligible. This has been corroborated by reaction-diffusion models which generate highly regular patterns in idealized homogeneous environments. However, such model-generated patterns are considerably more regular than natural patterns, which indicates that the concept of autogenous pattern formation is incomplete. Models can generate patterns which appear more natural when they incorporate exogenous random spatial heterogeneities (noise), such as microtopography or spatially varying soil properties. However, the mechanism through which noise influences the pattern formation has not been explained so far. Recalling that irregular patterns can form through stochastic processes, we propose that regular patterns can form through stochastic processes as well, where spatial noise is filtered through scale-dependent biophysical feedbacks. First, we demonstrate that the pattern formation in nonlinear reaction-diffusion models is highly sensitive to noise. We then propose simple stochastic processes which can explain why and how random exogenous heterogeneity influences the formation of regular and irregular patterns. Finally, we derive linear filters which reproduce the spatial structure and visual appearance of natural patterns well. Our work contributes to a more holistic understanding of spatial pattern formation in self-organizing ecosystems.

## Introduction

Life is the remarkable result of self-organization that lets complex organisms emerge from little seeds. The understanding of self-organization has made great progress since Alan Turing showed how it can be described by a set of reaction-diffusion equations [1,2]. This mechanism does not always lead to complex forms but sometimes conspicuously regular patterns, like the stripes on a tiger or the spots on a leopard [3]. Similar spatial patterns form in a wide range

**Data availability statement:** The source code is available at https://github.com/karlkastner/environmental-spatial-pattern-formation. A packed archive of the source code including

dependencies is available at https://zenodo.org/records/15147996.

**Funding:** The author(s) received no specific funding for this work.

**Competing interests:** None of the authors has any competing interests.

of ecosystems [4]. The patterns in these ecosystems appear strikingly similar, even though the biophysical processes driving the pattern formation strongly differ.

Environmental spatial patterns consist of patches with high biomass that alternate with bare ground, and can take various forms, depending on the environmental conditions [5,6]. They can be regular, i.e., consist of patches with similar size spaced in similar intervals corresponding to a characteristic length-scale, Fig 1 $b_i$, Fig 1$c_i$, or irregular, where the patch size and spacing varies strongly, without a finite characteristic length-scale, Fig 1 $a_i$. Patterns occur at the transition between two homogeneous states, for example, between that of a shrubland with continuous vegetation cover and that of barren land without any vegetation. Irregular patterns form when biophysical processes facilitate the growth or reduce mortality of established vegetation, but do not depend on the patch size. Examples are patterns forming through recurring fires [7] or grazing [8]. Irregular spatial patterns can be generated with simple reaction-diffusion models incorporating spatial noise [7–12].

Regular patterns form when there is both short-range facilitation and long-range competition, i.e., when feedbacks are scale-dependent [13]. An example for scale-dependent feedback is the flow of water from bare areas to vegetated patches in arid ecosystems. Regular patterns have a characteristic wavelength. The average distance between patches corresponds to the characteristic wavelength, and the average diameter of the patches is proportional to the characteristic wavelength.

The structure of regular patterns differs depending on the slope of the terrain: On flat terrain, patterns consist of vegetation patches scattered on bare ground when the water availability is low, Fig 1$b_i$, or gaps of bare ground interspersed in the vegetation, when the water availability is high. For an intermediate availability of water, neighbouring patches and spots connect to form labyrinthine patterns. Patterns forming on flat terrain are isotropic, as neither the arrangement nor the shape of the patches has a dominant direction. On hillslopes, patterns are anisotropic, consisting of vegetation stripes running parallel to the elevation contours, Fig 1$c_i$. Patterns forming on hillslopes are anisotropic, as the stripes are aligned in a particular direction.

Reaction diffusion-models with two or more coupled equations can generate regular patterns [6,14–18]. In arid ecosystems, the reaction terms accounts a.o. for the growth and dieback of vegetation, while the diffusion term accounts for the omnidirectional spread of water and vegetation on a level plane [17], while higher infiltration rates beneath vegetation facilitate the flow of water from bare ground towards vegetated patches [19]. Models for anisotropic patterns include an advection term accounting for the unidirectional flow of water [6,20–23]. The conditions for autogenous pattern formation and the characteristic wavelength can be determined by linear stability analysis [6,9,16,18,24–30]. Regular patterns form because the homogeneous bare or vegetated states are unstable, i.e., perturbations at a particular length scale grow over time. Regular patterns can form autogenously from an initially small perturbation in homogeneous environments, i.e., in the absence of exogenous spatial heterogeneities (noise). Patterns generated by models assuming homogeneous environments tend to be highly regular. The patches are of equal size and are spaced at an equal distance. The pattern thus repeats in a fixed interval, i.e., is periodic.

Natural regular patterns have also been classified as periodic, i.e., to repeat at their characterized wavelength. Differences between periodic and natural patterns have been attributed to superimposed noise, i.e., noise that is not an integral part of the pattern which can for example originate from image acquisition [31–33]. However, there is a systematic difference between periodic and natural regular patterns, as the patch size and spacing varies, Fig 1$b_i$, Fig 1$c_i$, and the frequency components are scattered over a wide frequency range,

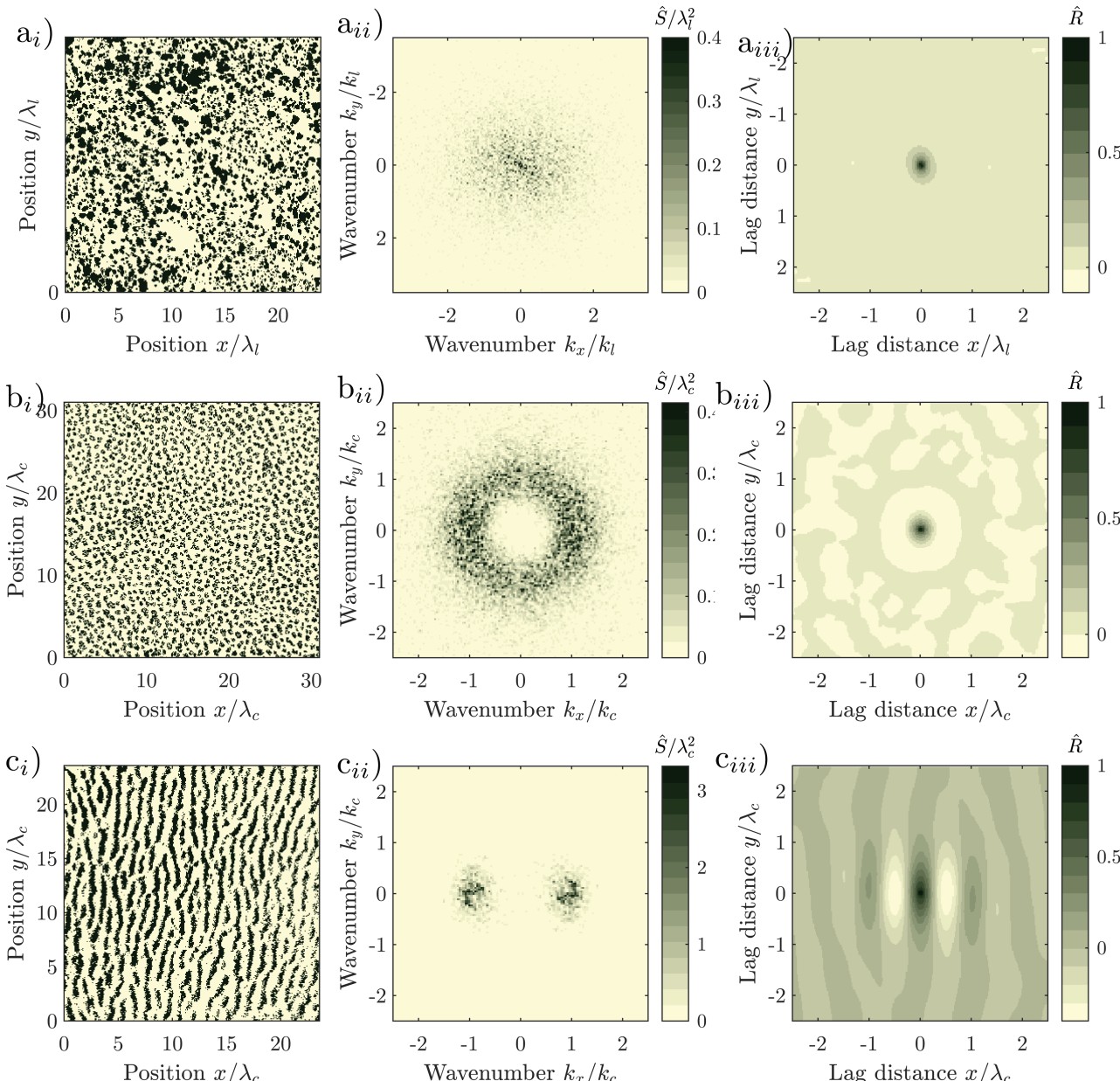

**Fig 1. Natural vegetation patterns.** ($a_i$) Irregular vegetation pattern, regularity $S_{rc}/\lambda_c = 0$, $p$-value of periodicity test 0.97, ($a_{ii}$) its periodogram, and ($a_{iii}$) its correlogram. Axes normalized by the cut-off wavelength $\lambda_l = 9$ m, beyond which components with shorter wavelength are suppressed. (44°10'10"N 5°16'22"E, Mt. Ventoux, France). ($b_i$) Isotropic regular vegetation pattern, regularity $S_{rc}/\lambda_c = 1.09$, $p$-value of periodicity test 0.98, $b_{ii}$) its periodogram and $b_{iii}$) its autocorrelation. Axes normalized by the characteristic wavelength $\lambda_c = 42$ m. (38°23'09"N 114°52'00"W, Lincoln County, Nevada, USA) ($c_i$) Anisotropic regular vegetation pattern regularity $S_{x+c}/\lambda_c = 1.3$, $S_{y+c}/\lambda_c = 3.4$, $p$-value of periodicity test 0.15, ($c_{ii}$) its periodogram, and ($c_{iii}$) its autocorrelation. Axes normalized by the characteristic wavelength $\lambda_c = 100$ m. (11°10'18"N 28°15'51"E, Kordofan, Sudan). Spurious low-frequency components have been suppressed in the periodograms. Images extracted from Google Maps 2023 Maxar Technologies imagery, and were processed in greyscale and thresholded for display. Patterns taken from the dataset of [34]. Note that throughout this manuscript, we use the terms "periodogram" and "correlogram" to the frequency spectrum and sample autocorrelation of an individual pattern, while we use the terms "spectral density" and "autocorrelation" for the frequency spectrum and autocorrelation of the underlying system or model.

Fig 1b$_{ii}$, Fig 1c$_{ii}$. This variation is intrinsic to the patterns and cannot be explained by super-imposing noise. This was first noticed by Kastner et al. [34]. They supported their findings by developing a statistical test not limited to the assumption of added noise, and applying it to a large global dataset of natural regular patterns. Kastner et al. [35] further introduced a method for quantifying the degree of regularity of spatial patterns, and showed in a metastudy that patterns generated with models emulating homogeneous environments are considerably more regular than natural patterns. However, it has not yet been explained why natural regular patterns systematically differ from periodic patterns. This lack of understanding impedes the accurate interpretation of environmental patterns as ecosystem resilience indicators [36–39].

Here, we propose that random exogenous heterogeneities influence the pattern formation itself, resulting in patterns with intermediate regularity. This is relevant, as noise can influence the stability of nonlinear systems [40]. There are many different sources of noise, i.e., random fluctuations in time and space, for example, spatially varying soil properties [41], microtopography [42], the random dispersal of seeds [43–46], the random local removal of vegetation or perturbation of soil through animals or humans [47–49] or the spatio-temporal variation of precipitation [50]. Time-independent spatial noise is a form of exogenous random heterogeneity which is also referred to as geodiversity [51].

Several studies show that model-generated patterns appear more similar to natural ones when the environment is heterogeneous [42,51–56]. However, these studies neither systematically explore the spatial structure of patterns nor do they explain why and how random exogenous heterogeneities influence it. The formation of irregular patterns has been described as a stochastic process [5,57]. Correspondingly, regular patterns can be interpreted as random fields generated by stochastic processes [35]. Stochastic in the sense that the pattern forming system can be described with a partial differential equation where the coefficients randomly vary in space [58,59], random in a sense that the fraction of the spatial variance contributed by each frequency component to a pattern is random. In consequence, every pattern is unique, even if it forms under statistically identical conditions as other patterns. However, the spatial structure of patterns forming under the same statistical conditions appears still similar as it is determined by spectral density of the stochastic process.

The stochastic process imposes the spatial correlation structure through filtering exogenous spatial heterogeneity. Kastner et al. [35] demonstrated this by synthesizing regular patterns appearing similar to natural ones through filtering spatial noise. Filtering retains components of the noise close to the characteristic wavelength of the system and attenuates components of the noise at length scales that are much smaller or larger. However, they fell short of showing that systems with scale-dependent feedbacks filter exogenous heterogeneities and did not elaborate on the stochastic aspects of pattern formation.

Here, we show that scale-dependent feedbacks filter random spatial exogenous heterogeneities, i.e., that patterns form through a stochastic process. We further show that filtering can explain the spatial structure of spatial regular isotropic vegetation patterns, and in particular the difference between natural patterns and periodic patterns generated with models assuming homogeneous environments. In the first section, we present our general idea how random exogenous heterogeneities influence the pattern formation. In the second section, we recall two nonlinear models of pattern formation, and explain how we introduce exogenous spatial heterogeneity to them. In the fourth section, we explore how irregular patterns form, and in the fifth and sixth section, we explore how isotropic and anisotropic regular patterns form.

## Pattern formation through filtering of exogenous heterogeneities

We consider ecosystems where spatial vegetation patterns form over time through the repeated interaction of biophysical processes and perturbation by exogenous heterogeneities. When the vegetation can be well approximated by a biomass density which changes continuously in space and time, then the system can be described by a partial differential equation:

$$\frac{\partial z}{\partial t} = \mathcal{A}(z, e). \tag{1}$$

where $z(t,x)$ is the system state which contains the relevant quantities which interact during the pattern formation. For systems forming irregular vegetation patterns, $z$ typically comprises only of the biomass. For systems forming regular vegetation patterns, $z$ typically comprises in addition to the biomass also of the the soil water content $w$ and surface water depth $h$, i.e., $z = (b, w, h)^T$. $e$ is a parameter which is spatially heterogeneous, such as spatially varying soil properties. $\mathcal{A}$ is a differential operator describing the change and interaction of the system state through biophysical processes. The pattern can be interpreted to form through successive filtering by the biophysical processes and perturbation by the random exogenous heterogeneity. Filtering makes the patterns more regular by suppressing and amplifying spatial components depending on their wavelength, while the random perturbation makes the pattern less regular. The filtering can result in a regular pattern where the biomass alternates with bare ground at a characteristic wavelength, even when the exogenous heterogeneity does not have a dominant wavelength. The system is deterministic when the parameter $e$ is changing deterministically in time and space or remains constant, i.e., when it is homogeneous. When the parameter $e$ changes randomly in time or space, then the pattern formation resembles a stochastic process [58].

Under certain conditions the system can approach an equilibrium. These conditions are given when the system is time-invariant and isotropic. Time-invariance requires that the environmental conditions including the exogenous heterogeneity remain constant over time. This requires that intermittent influences, such as precipitation, are represented by their long-time average. At the equilibrium $z_\infty$ the perturbation and filtering balance each other and the pattern remains stationary:

$$0 = \mathcal{A}(z_\infty, e). \tag{2}$$

As the pattern forms through repeated filtering, it is also influenced by the initial condition beside the exogenous heterogeneity [39]. However, as the exogenous spatial heterogeneity persistently influences the pattern formation at any time, its influence is usually much stronger than that of the initial condition.

## Linearization

To understand how random exogenous heterogeneities influence the spatial structure of vegetation patterns, we expand the system (1) as a series at the state $z_*$ and parameter $e_*$ and keep only the constant and linear terms:

$$\frac{\partial z_{\mathcal{L}}}{\partial t} = \mathcal{A}(z_*, e_*) + \frac{\partial \mathcal{A}}{\partial z}(z_{\mathcal{L}} - z_*) + \frac{\partial \mathcal{A}}{\partial e}(e - e_*). \tag{3}$$

Patterns form from an initially homogeneous vegetation cover under suitable conditions through the growth of frequency components close to the characteristic frequency of the

system. Here, we are interested in the spatial structure and corresponding spectral density at the stationary state, c.f. Eq 2. We do not consider the necessary conditions for pattern formation here, i.e., the stability of an initially homogeneous state. When the system is linearized close to a stationary state, then the state $z_{\mathcal{L}}$ of the linear system approaches the limit state $z_{\mathcal{L}\infty}$:

$$z_{\mathcal{L}\infty} = z_* - \left(\frac{\partial \mathcal{A}}{\partial z}\right)^{-1}\left(\mathcal{A}(z_*, e_*) + \frac{\partial \mathcal{A}}{\partial e}(e - e_*)\right). \qquad (4)$$

We simplify the notation by collecting the constant terms into $\tilde{z}_* = z_* - \left(\frac{\partial \mathcal{A}}{\partial z}\right)^{-1}$ $\left(\mathcal{A}(z_*, e_*) + \frac{\partial \mathcal{A}}{\partial e}e_*\right)$ and combine the operator into $\mathcal{L}^{-1} = \left(\frac{\partial \mathcal{A}}{\partial z}\right)^{-1}\frac{\partial \mathcal{A}}{\partial e}$:

$$z_{\mathcal{L}\infty} = \tilde{z}_* - \mathcal{L}^{-1}e. \qquad (5)$$

As the system is linear, each state variable can be determined independently of the other state variables, i.e., the biomass $b_{\mathcal{L}}$ can be determined as:

$$b_{\mathcal{L}\infty} = \tilde{b}_* - (\mathcal{L}^{-1})_b e. \qquad (6)$$

where $(\mathcal{L}^{-1})_b$ is the part of the system $\mathcal{L}^{-1}$ determining the biomass.

Here, we linearize around a homogeneous state $z_*$ with homogeneous parameter $e_*$ resulting in a space-invariant linear operator $\mathcal{L}^{-1}$. Homogeneous and space-invariant meaning that the values are constant in space. The spatial variance of the pattern $b_{\mathcal{L}\infty}$ thus entirely origins from the exogenous heterogeneity $e - e_*$.

We linearize the nonlinear operator $\mathcal{A}$ in a statistical sense, as a direct linearization at a single state, in particular at a homogeneous state, does not yield a stable linear operator. There always exists a stable linear operator $(\mathcal{L}^{-1})_b$ which exactly reproduces the spectral density and hence autocorrelation structure of the pattern $b$ generated by the nonlinear system, see further below. A suitable linear operator can be found as the weighted average of the linearization of $\mathcal{A}$ at several linearly independent homogeneous states or through the autocorrelation function and spectral density of a pattern generated by the nonlinear system.

In the remainder of the section, we shortly explain how isotropic patterns can form through filtering and how a linear filter can be determined from a spatial pattern. We elaborate this in greater detail in the supplement, section 1.

### Filtering in the spatial domain

The spatial heterogeneity $e$ can be filtered in the spatial domain by convolving ($\circledast$) it with the impulse response $\mathcal{IR}_b$ of the linear filter, Fig 2, c.f. ch. 3.3 in [60]:

$$b_{\mathcal{L}\infty} = \tilde{b} + \mathcal{IR}_b \circledast e. \qquad (7)$$

where $\tilde{b}$ is spatially homogeneous, as before.

Early models of spatial pattern formation were based on convolution [61,62]. The impulse response is the fundamental solution of the linear system, and is the response of the linear system to a single infinitesimal perturbation in an otherwise homogeneous environment, such as a point in space with higher infiltration. For the discretized system, the rows of the discrete operator $(L^{-1})_b$ are identical to the discrete impulse response $IR_{\mathcal{L}_b}$, shifted in location. The impulse response determines a region near the perturbation where vegetation growth is facilitated. In the presence of scale-dependent feedbacks, the impulse response has another region

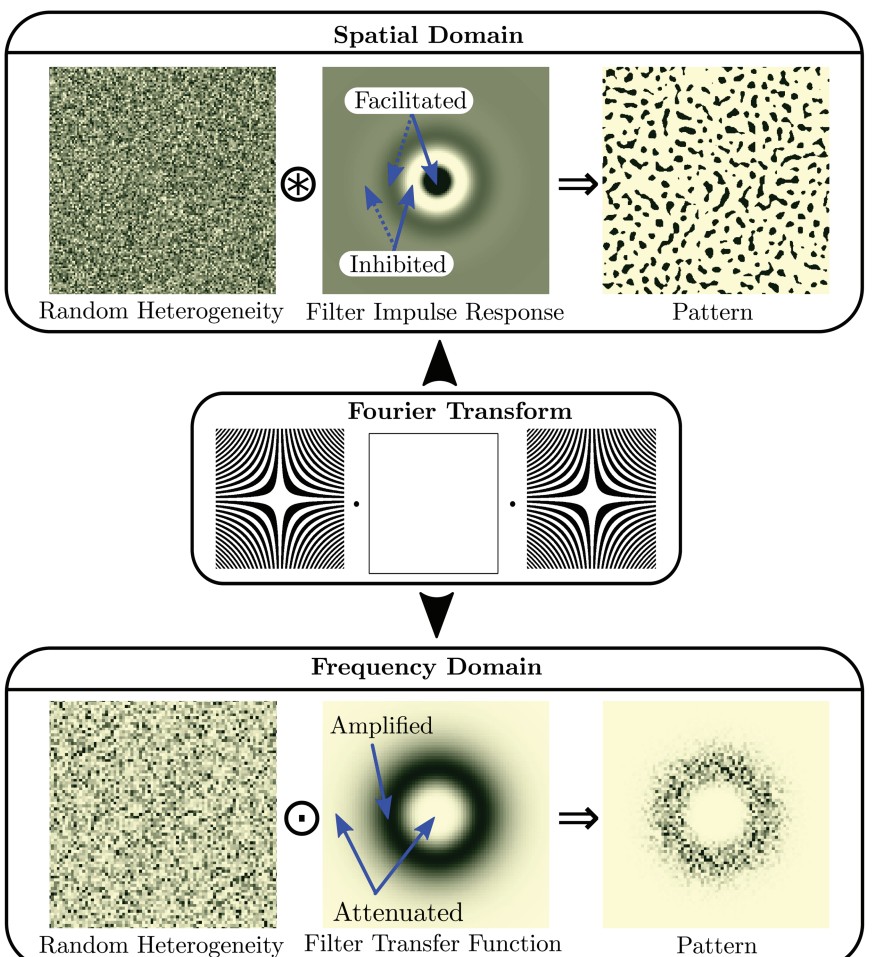

**Fig 2. Formation of an isotropic regular pattern by a stochastic process which filters a random spatial heterogeneity.** In the real domain, the random heterogeneity $e$ is filtered by convolving ⊛ it with the impulse response $\mathcal{IR}$ of the filter. Filtering amplifies spatial components of the exogenous heterogeneity close to the characteristic length scale where the autocorrelation function has its fist maximum and attenuates spatial components with shorter or longer wavelength. In the frequency domain, the Fourier transform $\mathcal{F}(e)$ of the random heterogeneity is filtered by multiplying it element-wise ⊙ with the transfer function $T$ of the filter, filtering amplifies frequency components of the noise close to the characteristic frequency where the spectral density $\mathcal{S} = |\mathcal{T}|^2$ is large, and attenuates components with higher or lower frequency.

further away where the growth of vegetation is inhibited. Regions of facilitation and inhibition alternate at an increasing distance from the origin at intervals corresponding to a patterns wavelength, c.f. Fig 2. When the environment is (randomly) perturbed in multiple places, the pattern results from the combined response of the system to the individual perturbations. When the environment is perturbed everywhere, the pattern is the convolution of the spatial heterogeneity map with the impulse response.

## Filtering in the frequency domain

The spatial heterogeneity $e$ can equivalently be filtered in the frequency domain, by multiplying the Fourier transform $\mathcal{F}(e)$ of the heterogeneity with the transfer function $(\mathcal{T}_\mathcal{L})_b$ of the linear filter:

$$b_{\mathcal{L}\,\infty} = \tilde{b} + \mathcal{F}^{-1}((\mathcal{T}_{\mathcal{L}})_b \cdot \mathcal{F}(e)). \tag{8}$$

where $\odot$ indicates pointwise multiplication. The transfer function determines the frequency response of the system, i.e., how each frequency component of the pattern is related to the corresponding frequency component of the exogenous heterogeneity. In the case of a mathematical model, the transfer function is the Fourier transform of the model equation and thus just an alternative representation of it, c.f. in ch. 3.5 in [60]. A filter amplifies components of the noise in the frequency range where its spectral density $(\mathcal{S})_{\mathcal{L}})_b$ is large and attenuates components in the frequency range where the spectral density is small, c.f. Fig 2. The spectral density is the squared magnitude of the transfer function:

$$(\mathcal{S}_{\mathcal{L}})_b = s^{-2}|(\mathcal{T}_{\mathcal{L}})_b|^2. \tag{9}$$

where $s$ is a scalar normalizing the volume of the spectral density to 1. In case of irregular patterns, the filter attenuates high frequency components, while for regular patterns, the filter attenuates frequency components that are both larger and smaller than the characteristic wavenumber of the system.

The spectral density $\mathcal{S}_b$ of a pattern is the product of both the spectral density of the filter $(\mathcal{S}_{\mathcal{L}})_b$ and of the exogenous heterogeneity $S_e$ and the system:

$$\mathcal{S}_b = (\mathcal{S}_{\mathcal{L}})_b \cdot \mathcal{S}_e. \tag{10}$$

The spectral density is the Fourier transform of the autocorrelation function $R_b$:

$$\mathcal{R}_b = \mathcal{F}(\mathcal{S}_b). \tag{11}$$

Here, we define the spatial structure as the autocorrelation function. The spatial structure is thus uniquely determined by the spectral density.

We focus our analysis on the spectral density $\mathcal{S} = s^{-2}|\mathcal{T}|^2$ of the filter without explicitly considering its phase shift $\arg(\mathcal{T})$. The reason is that the phase shift of isotropic filters is zero. It is also not insightful to analyze the phase of the frequency components of a pattern, as the linearly predicted frequency components are identical to that of the corresponding frequency components of the exogenous heterogeneity, which is random in our case. Isotropic systems have a zero phase shift as the patches are neither aligned, nor shaped, nor shifted in a preferential direction. This can be illustrated through tiling a pattern. The magnitude and phase of the frequency components within each tile is random. However, the expected value of the magnitude is the spectral density, which therefore can be estimated by averaging the periodogram of the tiles. In contrast, the expected value of the phase is undefined, as the decorrelates with increasing distance between tiles.

## Indentifying a linear filter from the spectral density

For any pattern, it is possible to find a linear system which generates similar patterns with the same spatial structure. The linear filter can be identified either by finding its linear operator $L_b$, its impulse response $(\mathcal{IR}_{\mathcal{L}})_b$ or its transfer function $(\mathcal{T}_{\mathcal{L}})_b$. The transfer function can be determined from the cross-spectral density $\mathcal{S}_{eb} = \mathcal{F}(e)^* \cdot \mathcal{F}(b)$ between the pattern and the spatial heterogeneity map [63],

$$(\mathcal{T}_\mathcal{L})_b = \mathcal{S}_{eb} \div \mathcal{S}_e. \tag{12a}$$

$$(\mathcal{IR}_\mathcal{L})_b = \mathcal{F}^{-1}((\mathcal{T}_\mathcal{L})_b). \tag{12b}$$

Here, $^*$ denotes the complex conjugate, $\cdot$ and $\div$ element-wise multiplication and division. The division requires that the spectral density of the noise is nonzero at all wavenumbers, which is the case for common noise spectra, except for the mean is not part of the spatial structure. The transfer function can be determined in this way both for linear and nonlinear systems. In the case of a nonlinear system, the transfer function describes the part of the pattern which can be predicted by a linear filter.

For isotropic patterns, the transfer function is real and positive so that the phase shift is zero. The transfer function can therfore be approximated as:

$$(\mathcal{T}_\mathcal{L})_b = s\sqrt{\mathcal{S}_b \div \mathcal{S}_e}. \tag{13a}$$

When the system is linear, the approximation is identical to the definition in Eq 12a. When the system is nonlinear, the approximation overestimates the magnitude of the linear transfer function for frequencies that have nonlinear contributions.

The linear filter $\mathcal{L}_b$ is thus always stable, as its impulse response $(\mathcal{IR}_\mathcal{L})_b$ can be interpreter as the autocorrelation $\widetilde{\mathcal{R}}$ of a stochastic process with spectral density $\widetilde{\mathcal{S}} = \sqrt{\mathcal{S}}$. Correlation matrices are definite for all non-zero frequency components, i.e., their eigenvalues are positive. The correlation matrix is indefinite for the mean, i.e., the corresponding eigenvalue is zero. The mean can be shifted into the stability region as the mean by definition does not affect the spatial correlation structure.

Systems with unidirectional flow generate anisotropic, i.e., striped patterns. Striped patterns often do not reach a stationary state, as they can migrate over time depending on the direction of the flow. The cross section of stripes is furthermore asymmetric along the direction of flow. For example, vegetation stripes in arid environments migrate uphill over time [65] and have a higher biomass concentration at the uphill side than at their downhill side. The phase of the transfer function is therefore not zero $\arg(T) \neq 0$ and the impulse response $\mathcal{IR}$ is not symmetric $\mathcal{IR}(x, y) \neq \mathcal{IR}(-x, -y)$. While anisotropic patterns typically do not reach a stationary state, they can approach a state where the spectral density does not change in time, i.e., where only the phase, but not the expected magnitude of the frequency components changes over time, i.e., $\partial S/\partial t = 0$ . We therefore can still find a linear filter which generates patterns with a similar spectral density and autocorrelation structure as the anisotropic pattern.

## Estimating of the spectral density

The spectral density $\mathcal{S}_b$ is in general not known. Though the discrete spectral density $S_b$ can be estimated from a discrete spatial map of a pattern. The spectral density $S_b$ is the expected relative magnitude of the periodogram $\hat{S}_b = |F(b)|^2$, i.e., the magnitude of the frequency components of a pattern. The spectral density can be either estimated by by smoothing the periodogram or by fitting a parametric density to the periodogram [34]. Here, we fit parametric densities by minimizing the Hellinger distance of the radial density i.e., the squared difference between the square root of the spectral density of the linear filter and the square root of the empirical density $\bar{S}$ of the pattern, c.f. [64]. The estimation of the transfer

function also requires the scale $s$ and the discrete spectral density $S_e$ of the heterogeneity $e$. The scale $s$ determines the spatial variance and can be set to an arbitrary value as the variance by definition does not influence the spatial correlation structure. We can generate patterns with similar spatial structure as a natural pattern, as long as the spectral density of the noise when generating the pattern is the same as that used for estimating the filter. The spectral density of the exogenous heterogeneity is usually not known for natural patterns. However, we can assume a reasonable spectral density. The simplest choice is the flat spectrum where the density is constant for all frequency components. Here, we model the heterogeneity as a first-order autocorrelated process, with exponentially decaying autocorrelation, see further below.

## Differences between nonlinear and linear filtering

Linear filters cannot predict components originating from the nonlinear interaction, which are an integral part of pattern forming systems. In particular, linear filters cannot predict the bimodality of the biomass distribution, and the related sharp transition between vegetated patches and bare ground or the fraction of the ground covered by vegetation. These properties are associated with high frequency components originating from nonlinear interaction between the frequency components. However, we will later see that linear filters can predict most of the frequency components of patterns forming in heterogeneous environments, and that the sharp transition between vegetated patches and bare ground can be reproduced by thresholding a pattern predicted by a linear filter.

We assess how well a linear filter can predict the frequency components of an isotropic pattern based on the spectral coherence $c_{eb}$ [66]:

$$c_{eb} = \frac{|S_{eb}|^2}{S_b \, S_e}. \tag{14a}$$

The value of $c_{eb}$ is 1 for frequency components that can be fully predicted by a linear filter and zero for frequency components that cannot be predicted from a linear filter at all. The spectral coherence accounts both for the predictability of phase and amplitude.

We assess the goodness of fit for the total spectrum by the weighted average $\bar{c}$ of the spectral coherence:

$$\bar{c}_{eb} = \int_{-\infty}^{+\infty} \int_{-\infty}^{+\infty} S_{xy} \, c \, \mathrm{d}k_x \, \mathrm{d}k_y = \frac{1}{\int_0^\infty k_r \, S_r \, \mathrm{d}k_r} \int_0^\infty k_r \, S_r \, c_r \, \mathrm{d}k. \tag{15}$$

We further evaluate the goodness of fit between the pattern predicted by the linear filter and the pattern generated by the grazing or the Rietkerk model after thresholding.

The frequency components generated by nonlinear interaction are mutually dependent, in contrast to the frequency components that can be predicted by the linear filter. However, while mutually correlated, the phases of the nonlinear components are still random, as they originate from the higher order terms of the series expansion in Eq 3 and can be expressed as infinite weighted sums of the phases of the random heterogeneity.

## Fraction of ground covered by vegetation

A generic pattern with unimodal biomass distribution generated by the linear filter can be transformed into a pattern with bimodal biomass distribution, for example by thresholding.

This generates high frequency components with similar mutual dependence as the high frequency components generated by nonlinear interaction in the nonlinear system. Here, we obtain a pattern $b$ with mean biomass $\bar{b}$ and fraction of the ground covered by vegetation $p$, by thresholding the generic pattern $b_\mathcal{L}$ generated by the linear filter:

$$b = \begin{cases} \bar{b}/p, & b_\mathcal{L} > \text{quantile}(b_\mathcal{L}, p), \\ 0, & \text{otherwise.} \end{cases} \tag{16}$$

For isotropic patterns, the threshold determines if a pattern is spotted, labyrinthine or gapped. Thresholding twice with an upper and lower level yields ringed patterns, c.f. Fig 3a. The fraction of the ground covered by vegetation $p$, and hence the threshold, can be determined for any pattern, with a suitable algorithm [67]. For the Rietkerk model, $p$ is nearly linearly dependent on the water availability, Fig 3b.

## Assessing the effect of random exogenous heterogeneity

We quantify the influence of random exogenous heterogeneities based on a few characteristic properties of the spectral density, in particular the characteristic wavelength and the degree of regularity [35]. The characteristic wavelength determines the distance at which a pattern appears most similar to itself when being shifted in space. The degree of regularity determines how high the similarity is. The regularity of a pattern is determined by the relative magnitude of the random spatial heterogeneity, with respect to the tendency of the system to restore order, i.e., the relative strength of the scale-dependent feedbacks. In addition to estimating the regularity of patterns, we test the patterns for a periodic spatial structure [34].

For regular spatial patterns, the frequency components with the largest magnitude are densely scattered in the periodogram close to the characteristic wavenumber $k_c$. The spectral density $\mathcal{S}$, i.e., the expected squared magnitude of the frequency components, consists correspondingly of a distinct lobe with a maximum $\mathcal{S}_c$ at the characteristic wavenumber. We define the characteristic wavelength $\lambda_c$, by the location of the maximum of the lobe $\lambda_c = 2\pi/k_c$.

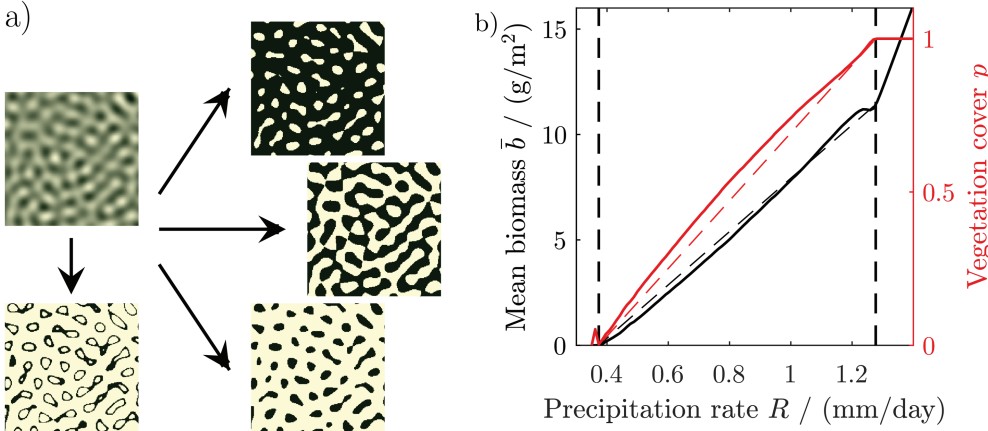

**Fig 3. Thresholding of filter-generated pattern.** (a) A generic isotropic pattern generated by a band-pass filter and a gapped, a labyrinthine and a spotted pattern obtained by thresholding the generic pattern so the vegetation cover is 80%, 50% and 20%, respectively, as well as a ringed pattern obtained by thresholding with a lower and upper level. (b) Relation between biomass and vegetation cover with precipitation in the Rietkerk model. Dashed lines indicate linear approximations.

A pattern appears the more regular, the higher and the narrower the lobe, i.e., the larger the ratio of the maximum $\mathcal{S}_c$ and the characteristic length scale $\lambda_c^d$ raised to the dimension $d$. The spectral densities of isotropic regular patterns have a lobe in the shape of a ring with a radius of the characteristic wavenumber $k_c$, Fig 1b$_{ii}$. We define the degree of regularity by the ratio $\mathcal{S}_{rc}/\lambda_c$, where $\mathcal{S}_{rc}$ is the maximum of the radial spectral density. The radial spectral density is determined by first transforming the spectral density into polar coordinates and then averaging it for every radius over all angles. The spectral density of irregular patterns has a lobe which is centred at the origin, i.e., the maximum of the spectral density occurs at the wavenumber $k_c = 0$, Fig 1a$_{ii}$. Irregular patterns therefore do not have a finite characteristic length scale and their regularity is zero. The spectral density of anisotropic patterns has a pair of lobes on the axis perpendicular to the stripes of the pattern with maxima at the characteristic wavenumber $\pm k_c$, Fig 1c$_{ii}$. We define the regularity along the primary axis as $\mathcal{S}_{x^+c}/\lambda_c$, where $\mathcal{S}_{x^+}$ is the spectral density averaged in the direction perpendicular to the stripes normalized over the positive half axis. We define the regularity along the secondary axis as $\mathcal{S}_{y^+c}/\lambda_c$, where $\mathcal{S}_{y^+}$ is the spectral density averaged in the direction perpendicular to the stripes and normalized over the half-axis. The regularity measures $\mathcal{S}_{rc}/\lambda_c$, $\mathcal{S}_{x^+c}/\lambda_c$ and $\mathcal{S}_{y^+c}/\lambda_c$ are scale-invariant. They are only determined by the spatial structure of the pattern. Their values remain the same irrespectively to which size the pattern is scaled, or in which units the length is measured.

## Models of pattern formation

We employ two nonlinear reaction-advection-diffusion models, to study how pattern forming systems respond to random exogenous heterogeneities, and how well this can be approximated by linear filters.

### Irregular patterns

We study irregular patterns with a spatially explicit grazing model [8,68]. This is a reaction-diffusion model in one variable, which evolves the biomass $b$ over time $t$ in two dimensional space $(x, y)$:

$$\frac{\partial b}{\partial t} = a\,b\left(1 - \frac{b}{k}\right) - c\frac{b^2}{b_n^2 + b^2} + e\left(\frac{\partial^2}{\partial x^2} + \frac{\partial^2}{\partial y^2}\right)b.$$

(17)

Table 1 states the model parameters.

### Regular patterns

Regular patterns can be modelled with reaction-diffusion models consisting of coupled equations with scale-dependent feedback. Here, we employ the Rietkerk model [17], which co-evolves vegetation biomass $b$, surface-water depth $h$ and soil-moisture $w$:

**Table 1. Parameters of the grazing model. Units guessed as they are not stated in the references.**

| Parameter | Unit | Value | Description |
|---|---|---|---|
| $c$ | $g/m^2/d$ | 2 | influences maximum grazing |
| $e$ | $m^2/d$ | 0.1 | diffusion rate |
| $a$ | $1/d$ | 1 | minimum regrowth rate |
| $b_n$ | $g/m^2$ | 1 | influences minimum grazing |
| $\bar{K}$ | $g/m^2$ | 8 | mean carrying capacity |

$$\frac{\partial b}{\partial t} = c_b\, U - d_b\, b + e_b \left( \frac{\partial^2}{\partial x^2} + \frac{\partial^2}{\partial y^2} \right) b, \tag{18a}$$

$$\frac{\partial w}{\partial t} = I - U - r\, w + e_w \left( \frac{\partial^2}{\partial x^2} + \frac{\partial^2}{\partial y^2} \right) w, \tag{18b}$$

$$\frac{\partial h}{\partial t} = R_h - I + v_h \frac{\partial h}{\partial x} + e_h \left( \frac{\partial^2}{\partial x^2} + \frac{\partial^2}{\partial y^2} \right) h, \tag{18c}$$

where $U$ is the water uptake by plants:

$$U = g_b \frac{w}{w + k_U} b, \tag{18d}$$

and $I$ the infiltration of water into the soil:

$$I = ah \frac{b + k_I\, w_0}{b + k_I}. \tag{18e}$$

In case of flat terrain, i.e., $v_h = 0$, surface water diffuses omnidirectionally, The model generates isotropic, i.e., spotted, labyrinthine or spotted patterns, depending on the water availability. In case of a hillslope, water flows downhill ($|v_h| > 0$) the model generates anisotropic, i.e., striped patterns. The model parameters are stated in Table 2.

## Model of random exogenous heterogeneity

Here, we limit our analysis to exogenous heterogeneity which is constant in time, i.e., spatial noise. Such heterogeneity can be considered to result from processes such as soil formation and erosion which progress on longer timescales than that of vegetation and do not have to be explicitly modelled. Exogenous heterogeneity thus has its own spatial correlation structure $\mathcal{R}_e$ and corresponding spectral density $\mathcal{S}_e$, which in turn influences the spatial correlation structure $\mathcal{R}_b$ and corresponding spectral density $\mathcal{S}_b$ of the vegetation pattern c.f. Eq 10.

**Table 2. Parameters of the Rietkerk model.**

| Symbol | Value | Unit | Parameter |
|---|---|---|---|
| $c_b$ | 10 | g mm$^{-1}$ m$^{-2}$ | conversion rate of water to biomass |
| $d_b$ | 0.25 | d$^{-1}$ | dieback rate of biomass |
| $g_b$ | 0.05 | mm g$^{-1}$ m$^{-2}$ d$^{-1}$ | uptake rate for nearly-dry soil |
| $e_b$ | 0.1 | m$^2$ d$^{-1}$ | diffusion of biomass |
| $\bar{a}$ | 0.2 | d$^{-1}$ | scales the maximum infiltration rate |
| $k_I$ | 5 | mm | half-saturation of water-uptake |
| $r_w$ | 0.2 | d$^{-1}$ | percolation rate of soil water |
| $e_w$ | 0.1 | m$^2$ d$^{-1}$ | diffusion rate of soil water |
| $v_h$ | varied | m d$^{-1}$ | velocity of surface water |
| $e_h$ | varied | m$^2$ d$^{-1}$ | diffusion rate of surface water |
| $R_h$ | varied | mm d$^{-1}$ | precipitation rate |
| $w_0$ | 0.2 | 1 | scales the minimum infiltration rate |
| $k_U$ | 5 | g m$^{-2}$ | half-saturation of water uptake |

Most studies accounting for exogenous spatial heterogeneity neglect the correlation structure and simply add uncorrelated white noise to a discrete model coefficient [8]. Here, we account for exogenous spatial heterogeneity by varying one coefficient of the respective reaction-advection-diffusion system according to a first-order autocorrelated spatial stochastic process. In particular, we vary the carrying capacity coefficient $K$ in the grazing model [8,68], and vary the infiltration coefficient $a$ in the Rietkerk model [17].

We model the spatially varying heterogeneity $e$ with mean $\mu_e$ and variance $\sigma_e^2$ as a log-normal stochastic process, Fig 4:

$$e = \exp\left(\varepsilon\right). \tag{19}$$

where $\varepsilon$ is a Gaussian process with mean $\mu_\varepsilon$, variance $\sigma_\varepsilon^2$ and isotropic first-order autocorrelation structure $\mathcal{R}_\varepsilon$:

$$\mu_\varepsilon = \ln\left(\mu_e\right) - \frac{1}{2}\ln\left(1 + \sigma_e^2/\mu_e^2\right), \tag{20a}$$

$$\sigma_\varepsilon^2 = \ln\left(1 + \sigma_e^2/\mu_e^2\right), \tag{20b}$$

$$\mathcal{R}_\varepsilon(r) = \exp\left(-r/\theta_\varepsilon\right), r = \sqrt{x^2 + y^2}. \tag{20c}$$

where $\theta_\varepsilon$ is the correlation length of $\varepsilon$ at which $R_\varepsilon\left(\theta_\varepsilon\right) = \exp\left(-1\right)$. The correlation structure $R_e$ of $e$ is close but slightly different to a decaying exponential, see supplement section 1.7 for details. The process resembles a geometric Ornstein-Uhlenbeck process in two dimensions and is equivalent to a first order low-pass filter. To separate the heterogeneity from that of the vegetation pattern, we chose correlation length of the heterogeneity that is much larger than that of the patches in case of the Rietkerk model, and much smaller than that of the patches in case of the grazing model. This corresponds to observations that natural vegetation patterns vary considerably at length scales larger than individual patches or stripes [34].

We report the degree of exogenous spatial heterogeneity relatively to the mean by the coefficient of variation cv($e$):

$$\text{cv}(e) = \frac{\sigma_e}{\mu_e}. \tag{21}$$

We keep the mean and the spectral density of the coefficient constant and only vary the degree of heterogeneity cv($e$).

The infiltration is a combination of exogenous and endogenous factors. In the Rietkerk model, the infiltration rate $I$ is the product of the infiltration coefficient $a$, a factor accounting for the enhancement of infiltration by the vegetation $ie = (b + k_I w_0)/(b + k_I)$, and the water depth $h$, c.f. Eq (18e). We compare the relative influence of the three factors on the infiltration through the following decomposition:

$$\frac{I - \bar{I}}{\sigma_I} = c_a \frac{a - \bar{a}}{\sigma_a} + c_{ie} \frac{ie - \overline{ie}}{\sigma_{ie}} + c_h \frac{h - \bar{h}}{\sigma_h}. \tag{22}$$

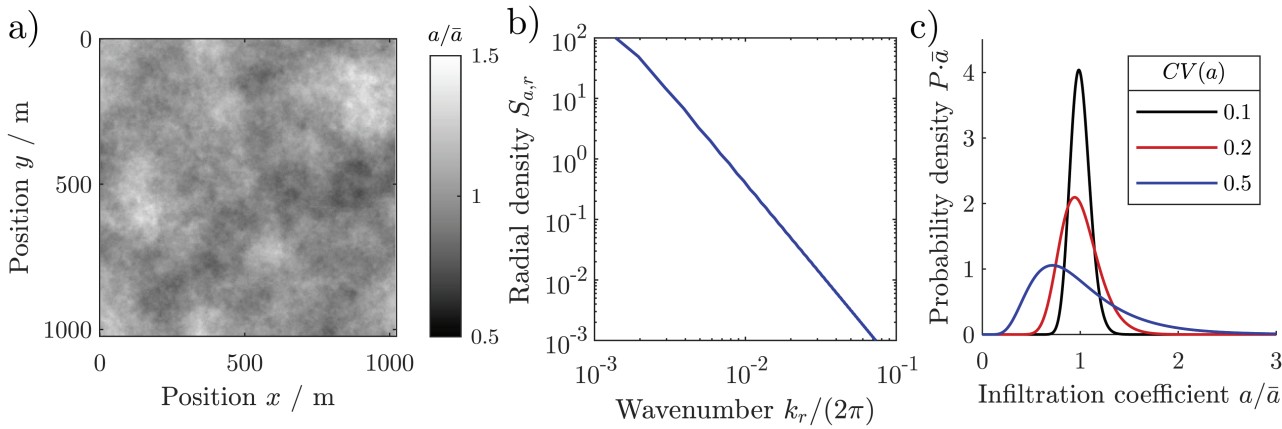

**Fig 4. Spatial heterogeneity of the infiltration coefficient $a$ with mean $\bar{a}$ and relative standard deviation cv($a$) at a unit grid cell size, (a) spatial map, (b) radial spectral density, (c) statistical distribution of the coefficients for three different degrees of heterogeneity cv($a$).**

$\bar{I}, \bar{a}, \bar{ie}, \bar{h}$ are the means, and $\sigma_I, \sigma_a, \sigma_{ie}, \sigma_h$ are the standard deviations. The coefficients $c_a, c_{ie}$ and $c_h$ measure the fraction of the heterogeneity of infiltration contributed by the respective explanatory variables. Their values are identical to the correlation coefficient, when the explanatory variables are independent.

The continuous process Eq 19 has to be approximated by a discrete process in the numerical model, as it has a finite spatial extent and finite number of grid cells, resulting both in a finite spectral and a finite spatial resolution. We account for the finite spatial resolution by averaging the autocorrelation of the continuous process over the area of a grid cell [59,69]. We account for the finite spatial extent by multiplying the autocorrelation with a smooth Fourier window. We generate the heterogeneity by convolving uncorrelated white noise with the correlation structure of the discrete process. Supplement section 1.7 provides details on generating the random exogenous heterogeneity.

## Formation of irregular patterns through low-pass filtering

We first revisit the formation of irregular patterns, as spatial noise has been recognized as essential for their formation, in contrast to regular patterns [7,9–12]. There are two different conceptual models for the spectral density and hence spatial structure of irregular patterns: The first model considers irregular patterns as scale free due to the lack of a finite characteristic length-scale, and describes the spectral density as a power-law [70,71], where the magnitude of frequency components decays inversely with increasing wavenumber, but diverges to infinity towards the zero wavenumber. Patterns with power law densities form through processes with long range dependence, as their autocorrelation decays slower than an exponential [72]. The second model considers irregular patterns to form through processes with short range dependence, where the change at any moment at any point only depends on the immediate neighbourhood of the point. This can be described by processes with an exponentially decaying autocorrelation, i.e., by processes with short range dependence [5,73,74]. The corresponding spectral density is that of a low-pass filter, where the magnitude of the frequency components decay inversely with increasing wavenumbers, but reach a finite maximum for low wavenumbers. Some studies alleged that irregular patterns can have both a power-law spectrum and an exponentially decaying autocorrelation [75]. However, this is inconsistent

with the theory of spectral analysis which relates the spectral density and autocorrelation as Fourier transform pairs: power-law densities transform into autocorrelation functions with long range dependence, and low-pass densities transform into autocorrelation functions short range dependence [76]. The disparity with the theory could indicate that irregular patterns form through non-stationary processes or through a combination of both short and long-range processes. Their analysis would require more complex models of the spectral density and autocorrelation. The suitable model likely depends on the scale at which patterns are analysed.

Here, we employ the short-range model, as it better fits both the natural and model-generated irregular patterns studied by us on plot-scale, and because the linearization of the grazing model yields a low-pass filter. A low-pass filter is characterized by its cut-off wavenumber $k_l$, beyond which frequency components with a higher wavenumber are suppressed. In the spatial domain, this corresponds to suppressing spatial structures smaller than the cut-off length $\lambda_l = 2\pi/k_l$. Here, we define the cut-off wavenumber as the wavenumber where the spectral density has dropped to 50% of its maximum value $S(k_l) = S(0)/2$, in accordance with the common definition in the field of signal processing ch. 3.7 in [60]. Our example of a natural irregular pattern consists of dwarf junipers, Fig 1a$_i$. Juniper patterns form because young (small) shrubs have a higher mortality than mature (large) shrubs, which can for example be explained by browsing [77], i.e., the pattern formation is not driven by the redistribution of water like the regular patterns. While no model for the formation of spatial juniper patterns has been published so far, patterns with similar frequency spectra can be generated with simple reaction-diffusion models [7,8]. Such models only generate patterns when they incorporate spatial noise.

Here, we use the spatially extended grazing model [8,68], and generate patterns by spatially varying the carrying capacity. We generate a sequence of patterns with the grazing model for different degrees of heterogeneity $0.01 \leq \text{cv}(K) \leq 1$. The patterns generated appear visually similar to natural irregular patterns, and have a similar spectral density, and corresponding correlation structure, Fig 5a$_i$-a$_{iii}$. The spectral coherence also indicates that a linear filter can predict the spatial structure to a large extent, with coherence values of 0.5 already for small degrees of exogenous heterogeneity (cv($K$) >0.01), and above 0.9 for high degrees (cv($K$) = 1), Fig 5d. Spectral coherence particularly high in the low frequency range, where the spectral energy of the pattern is concentrated, Fig 5c.

Motivated by our observation that the spectra of irregular patterns is similar to those of low-pass filter, and that the patterns generated by the grazing model can be predicted by a linear filter, we approximate the pattern formation by low-pass-filtering of the exogenous spatial heterogeneity. In particular, we choose a higher order low-pass filter with two parameters, which together determine the cut-off wavenumber and the rate of decay for frequency components at higher wavenumbers. We report here only the results and provide the details of the filter in the supplement, Section 2.1. The spectral density of the higher-order low-pass filter fits the spectral density of the natural pattern well, Fig 1a$_{iii}$, $R^2 = 0.95$. It also fits the spectral density of patterns generated with the grazing model well, with a goodness of fit of close to 1 independently of the degree of heterogeneity, Fig 5c.

Low-pass filtering does not only reproduce the spectral density and spatial structure of irregular patterns, but can even predict the location of individual patches in a particular pattern generated by nonlinear reaction-diffusion models. We demonstrate this by generating pairs of patterns from the same spatial map of randomly varying coefficients, one pattern of each pair with the nonlinear grazing model, and the other pattern with the linear low-pass filter. The patches of the paired patterns largely overlap, implying that the nonlinear

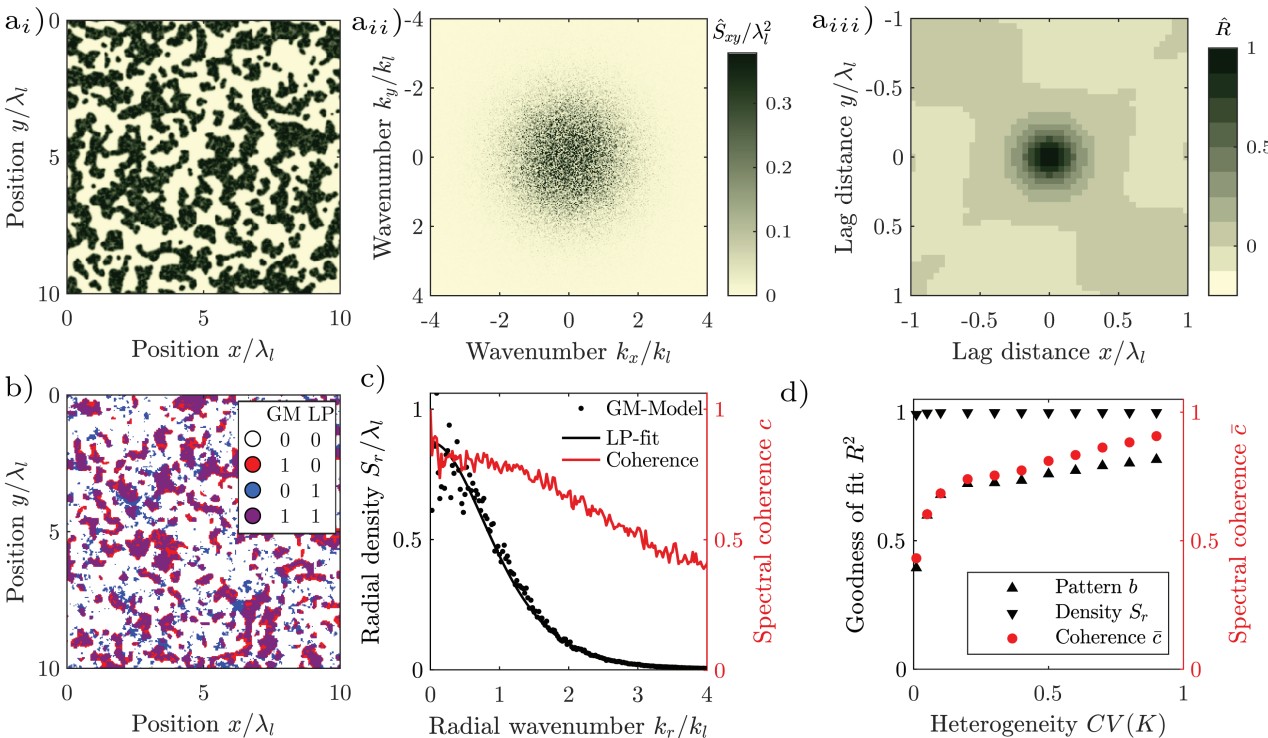

**Fig 5. Generic irregular patterns.** ($a_i$) Irregular pattern generated with the grazing model [8,68] (carrying capacity $K$ with mean $\mu_K = 8$, spatial coefficient of variation $cv(K) = 0.1$ and correlation length $\theta_K = 1$ m, maximum grazing rate $c = 2$, diffusion $d = 0.1$, initial homogeneous biomass density $b_0 = 2.4724$). Axes are normalized with the cut-off wavelength length $\lambda_l$. (b) periodogram and (c) correlogram of the pattern generated with the grazing model, Axes normalized with the cut-off length $l$ and corresponding wavenumber $k_l = 2\pi/\lambda_l$. (d) Overlap of the same pattern generated with the grazing model (GM) and with a low-pass filter (LP) from the same spatial map of the parameter $K$. Both patterns were thresholded for the comparison. The patches largely overlap and with most differences being located at patch fringes. (e) estimated radial density, fitted low-pass density and spectral coherence. (f) Average spectral coherence, goodness of fit of the low-pass generated patterns and of their spectral density with respect to that of the pattern generated by the grazing model, depending on the degree of spatial heterogeneity $cv(K)$.

grazing model generates spatial patterns by low-pass-filtering, which can be reasonably well approximated by the linear low-pass filter as long as the degree of exogenous heterogeneity is sufficiently high ($R^2 > 0.68$ for $cv(K) \leq 0.1$), Fig 5d. It is likely that natural irregular patterns form through similar stochastic processes, wherever diffusion suppresses high-frequency, i.e., small-scale, components introduced by spatial noise such as randomly varying microtopography or soil properties. The similarity of irregular pattern formation to filtering motivates us to study whether filtering of exogenous heterogeneities also plays a role in the formation of regular patterns.

## Isotropic regular patterns

Regular patterns can form in dryland ecosystems where redistribution of water drives the pattern formation [6,17]. When the terrain is flat, i.e., not sloping, the patterns are isotropic. Patterns are spotted, i.e., consist of isolated vegetation patches, when the water availability is relatively low. We illustrate the appearance of isotropic regular patterns with a spotted pattern from Nevada, USA, Fig 1$b_i$. The size of spots, the distance between spots and their shape varies, and the spots are not arranged in a regular grid. The correlogram has a local maximum at a lag-distance corresponding to the characteristic wavelength $\lambda_c$ between neighbouring

patches, but decays to zero for increasing lag-distances, Fig 1b$_{iii}$. The frequency components of the pattern are scattered in an annular region with characteristic wavenumber $k_c = 2\pi/\lambda_c$, Fig 1b$_{ii}$. The pattern has an intermediate degree of regularity ($S_{rc}/\lambda_c = 1.09$). The statistical test does not classify it as periodic ($p = 0.98$).

The Rietkerk model generates regular patterns autogenously in idealized homogeneous environments, i.e., in absence of exogenous spatial heterogeneity. We generated such a pattern, Fig 6a$_i$. The pattern is highly regular ($S_{rc}/\lambda_c = 7.8$), and classified as periodic by the statistical test ($p = 0.00$). The patches are of nearly identical size and spaced in a hexagonal grid. The distance between patches hardly varies, Fig 6a$_i$. The autocorrelation reflects the hexagonal grid-structure of the pattern, Fig 6a$_{iii}$. Correspondingly, the spectral density consists of six narrow peaks located on a ring with radial wavenumber $k_c$ in 60° intervals, Fig 6a$_{ii}$. The pattern generated in homogeneous environments ($cv(a) = 0$) is considerably more regular than the natural patterns, c.f. Fig 1b$_i$. The model therefore does not reproduce the variation of the patch size, and consequently neither reproduces the decay of the spatial autocorrelation nor the scatter of the periodogram which are typical for natural isotropic regular patterns.

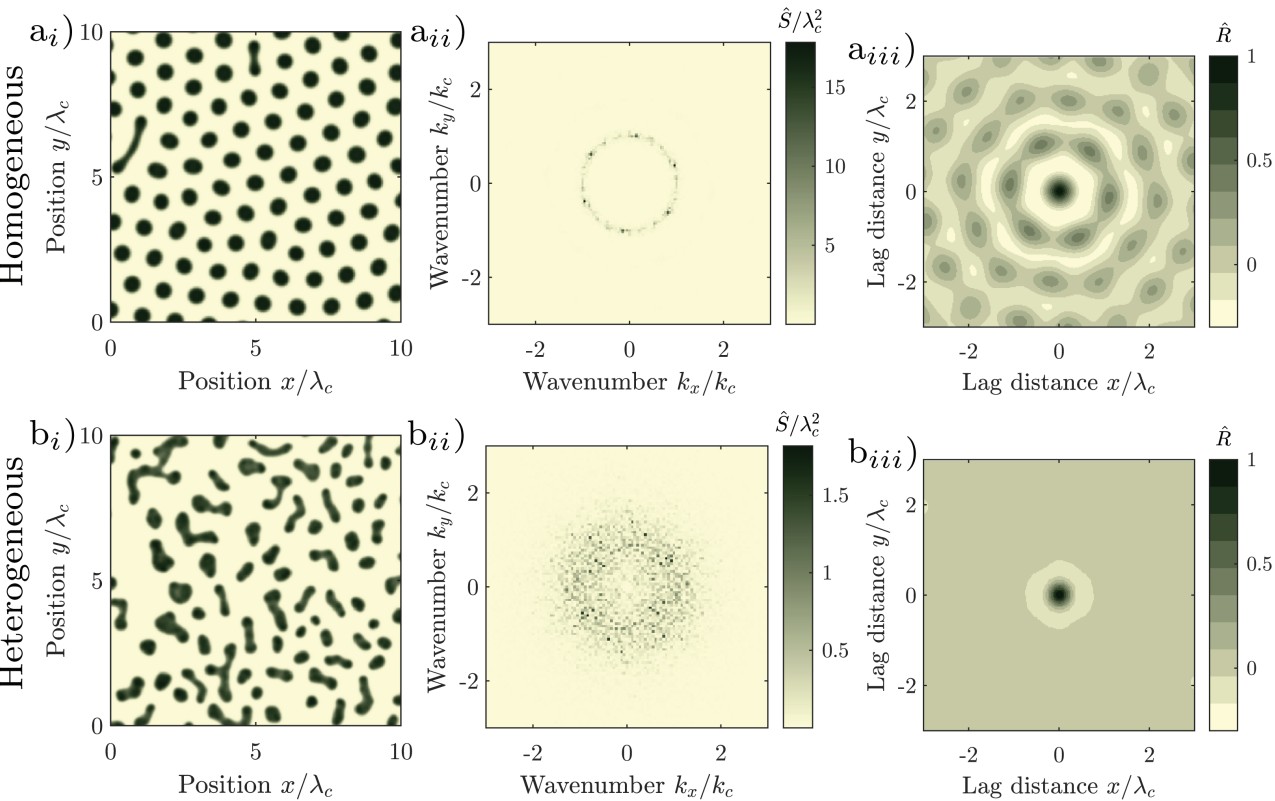

**Fig 6. Regular isotropic patterns generated with the Rietkerk model**. (rainfall intensity $R = 0.7$ mm/d, runoff velocity $v_x = 0$, water diffusion $e_x = e_y = 100$ m²/day, spatial extent $L^2 = (1024\,\text{m})^2$, spatial resolution $\Delta x = 1$ m, final time $T = 1369$ years, random initial condition). Other parameters have the same values as in [17]. (a$_i$) Highly regular pattern generated in a homogeneous environment ($cv(a) = 0$), regularity $S_{rc}/\lambda_c = 7.8$, $p$-value of periodicity test 0.00, (b$_i$) Pattern with intermediate regularity generated in a heterogeneous environment ($cv(a) = 0.3$). regularity $S_{rc}/\lambda_c = 0.95$, $p$-value of periodicity test 0.63. (a$_{ii}$,b$_{ii}$) corresponding periodograms, (a$_{iii}$,b$_{iii}$) corresponding correlogram. The patterns are cropped to an extent of 10 wavelengths for display, approximately one quarter of the model domain.

## Effect of exogenous heterogeneity on the formation of isotropic regular patterns

To reproduce the spatial structure of natural regular patterns, we spatially vary the infiltration coefficient $a$ in the Rietkerk model. Patterns generated in a spatially heterogeneous environment, Fig 6b$_i$ are indeed considerably more similar to natural patterns, Fig 1b$_i$, than patterns generated in a spatially homogeneous environment, Fig 6a$_i$. For an intermediate degree of exogenous random spatial heterogeneity ($cv(a) = 0.3$), the shape, size and distance between the patches varies, and patches are not arranged in a regular grid. The autocorrelation oscillates, but the oscillation is strongly damped, Fig 6b$_{iii}$. The frequency components are scattered in a wide region in a ring around the central wavenumber, Fig 6b$_{ii}$. While the pattern is less regular than in the homogeneous case, the pattern is still regular ($\mathcal{S}_{rc}/\lambda_c = 0.95$), as the lobe of the spectral density is well separated from zero ($k_c > 0$).

We systematically explore how the Rietkerk model responds to spatial noise, by generating a series of patterns with different degrees of spatial heterogeneity $cv(a)$ of the infiltration coefficient $a$. We repeat the experiment for three for three different levels of aridity, to explore the effect of the exogenous heterogeneity on spotted, labyrinthine and gapped patterns. In a spatially homogeneous environment, ($cv(a) = 0$) the Rietkerk model generates highly regular patterns, with values of $\mathcal{S}_{rc}\lambda_c$ of 4.9 for spotted, 3.8 for labyrinthine and 4.7 for gapped patterns, Fig 7a. The statistical test classifies the pattern as periodic ($p$=0.00). With increasing degree of exogenous heterogeneity, differences between the pattern types diminish and the regularity of the patterns decreases rapidly, similar to a power law with $\mathcal{S}_{rc}/\lambda_c \approx 0.3 cv(a)^{-0.5}$. The statistical test classifies the patterns as not periodic even when the degrees of exogenous heterogeneity is small ($cv(a) > 0.01$). The regularity decreases to a value of $\mathcal{S}_{rc}/\lambda_c$ of 0.44 when the degree of regularity is high ($cv(a) \leq 0.5$). The rapid decrease of the regularity during low degrees of heterogeneity is caused primarily by the rapid decrease of the maximum of the radial spectral density $S_{rc}$, while the more gradual decrease during high degrees of exogenous heterogeneity is caused by the gradual increase of the characteristic wavelength $\lambda_c$. The characteristic wavelength $\lambda_c$ is 44 m in homogeneous environments ($cv(a) = 0$), irrespectively of the pattern type, Fig 7b. The characteristic wavelength increases nearly linearly with the degree of heterogeneity and reaches values of 130 m for spotted, 132 for labyrinthine and 141 m for gapped patterns at $cv(a) = 0.5$.

In all experiments, the fraction of the total heterogeneity of infiltration stemming from the exogenous heterogeneity $c_a$ is small, never exceeding 18% in our experiments, Fig 7c. The infiltration feedback $ie$ explained almost all of the heterogeneity of heterogeneity, even in the experiments where the degree of exogenous heterogeneity is large. While the exogenous heterogeneity only slightly perturbs infiltration at any moment in time, its large effect on the spatial structure results from the integration of its small influence on the infiltration over time.

The trends of the regularity and wavelength are similar for spotted, labyrinthine and gapped patterns. This indicates that the spatial correlation structure is very similar between the pattern types and primarily determined by the exogenous heterogeneity. The spatial structure of the patterns is closer to that of periodic pattern when the degree of exogenous heterogeneity is relatively low, with the value of the regularity being closer to the limit value $L/\lambda_c$ of periodic patterns. The spatial structure of the patterns is closer to that of irregular patterns when the degree of exogenous heterogeneity is relatively high, with the regularity and wavelength being closer to the limit values $S_{rc}/\lambda_c = 0$ and $k_c = 2\pi/\lambda_c = 0$ of irregular patterns. While the water availability determines the fraction of ground covered by vegetation and hence the pattern type, it only slightly influences the spatial correlation structure of the

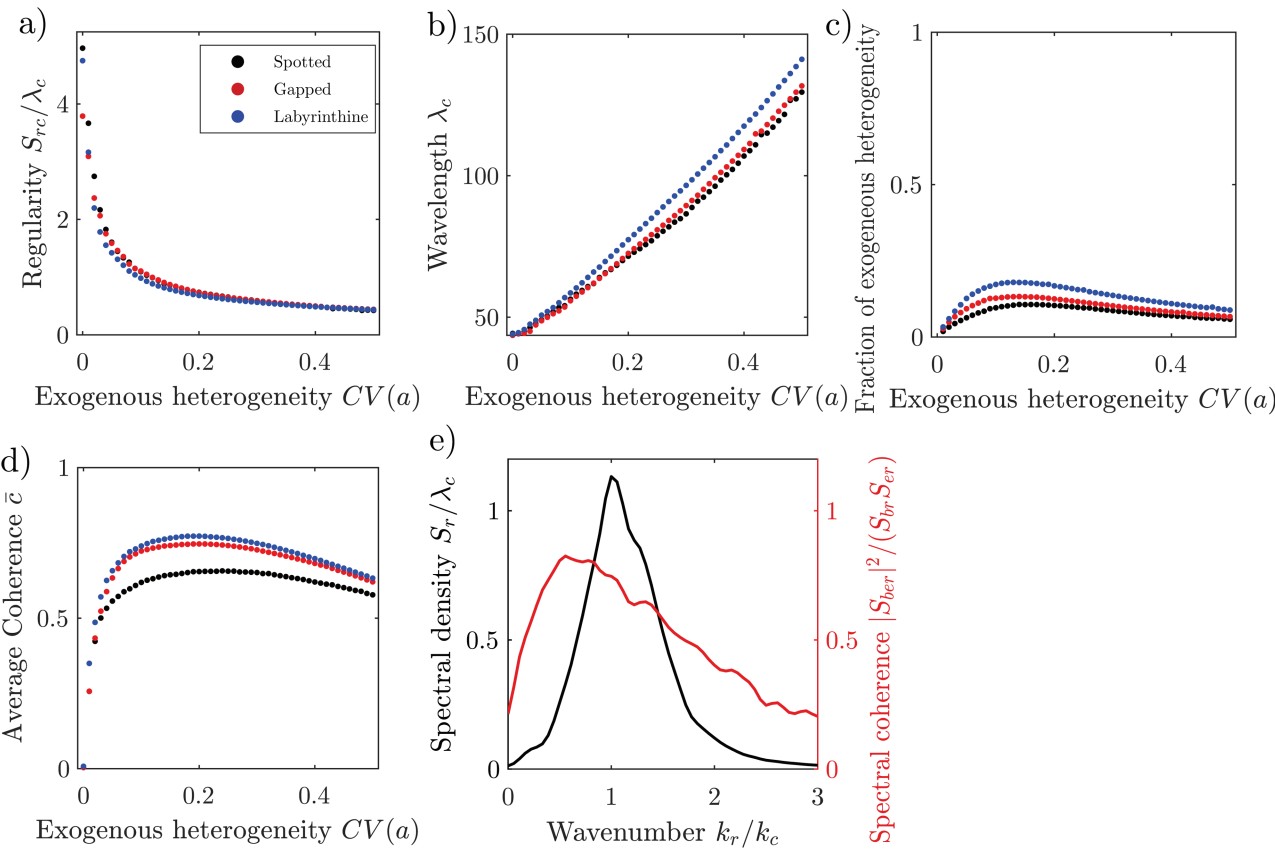

**Fig 7. Response of the Rietkerk model to environmental heterogeneity in flat terrain.** (a) Regularity and (b) wavelength of isotropic patterns generated with the Rietkerk model depending on the degree of exogenous spatial heterogeneity cv($a$) of the infiltration coefficient $a$. (c) Fraction of the total heterogeneity of infiltration contributed by the exogenous spatial variation of the infiltration coefficient $c_a$, c.f. Eq 22. (e) Spectral coherence depending on the degree of exogenous spatial heterogeneity. (f) Spectral density of a spotted pattern generated with the Rietkerk model (cv($a$) = 0.1) and spectral coherence with the exogenous spatial heterogeneity.

patterns. The similar response of all three pattern types to exogenous heterogeneity is expected, as the autocorrelation structure is by definition independent of the mean biomass, the primary discriminator of the pattern types. Random exogenous heterogeneity thus fundamentally changes the spatial structure of regular patterns, and does not just superimpose on a periodic pattern which would form in absence of noise, even if its magnitude is relatively small. This can be understood through filtering.

## Formation of isotropic regular patterns through band-pass filtering

The concentration of frequency components with large magnitude around the characteristic wavenumber suggests that the patterns can form through band-pass filtering where components with a wavelength close to the characteristic length-scale are amplified, while components with a smaller or a large wavelength, i.e., a lower or a higher wavenumber, are suppressed. Band-pass filtering occurs when two biophysical quantities, such as vegetation and water, interact but diffuse at different rates [9]. For vegetation patterns, this intuitively follows from the water redistribution: Vegetation establishes in places where water collects or can more easily infiltrate, such as in small depressions or cracks between rocks.

High-frequency perturbations at length scales much shorter than that of the pattern are smoothed out by diffusion, while low-frequent perturbations at length scales much longer than that of the pattern are ineffective as there is no flow over longer distances in dryland ecosystems where water is locally consumed by vegetation. Conversely, exogenous heterogeneities varying at a length scale close to that of the system form a template for the pattern which is amplified by feedbacks in the system.

We demonstrate that patterns can form through band-pass filtering of exogenous heterogeneities, by generating a one-dimensional pattern with the Rietkerk model where we slightly vary the infiltration coefficient $a$ sinusoidally in space with gradually varying wavelength, i.e., a short wavelength of $1/4\,\lambda_c$ at one end and $4\,\lambda_c$ at the other end of the modelling domain, where $\lambda_c$ is the characteristic wavelength of patterns generated by the model in an environment with a spatially homogeneous infiltration coefficient. The Rietkerk model responds differently depending on the wavelength of the perturbation of the infiltration coefficient $a$. The vegetation is strongly correlated with the infiltration coefficient, where the wavelength of the perturbation is close to the characteristic wavelength $\lambda_c$ of the pattern, Fig 8a. This is because the vegetation patches occupy the regions where the infiltration coefficient is high, Fig 8c. When the infiltration coefficient varies at a wavelength much larger or smaller than that of the pattern, then the vegetation patches cannot align with regions of higher infiltration, as the sizes are too dissimilar, Fig 8c, Fig 8d. Exogenous heterogeneities therefore form a template for the pattern, when they vary at a length scale similar to the characteristic wavelength of a system, but do not strongly influence the pattern when the length scale is much shorter or longer. When the environment varies randomly in space, then the pattern is imposed by the frequency components of the noise with length scales close to that of the system, i.e. the system filters the spatial components of the heterogeneity depending on their wavelength.

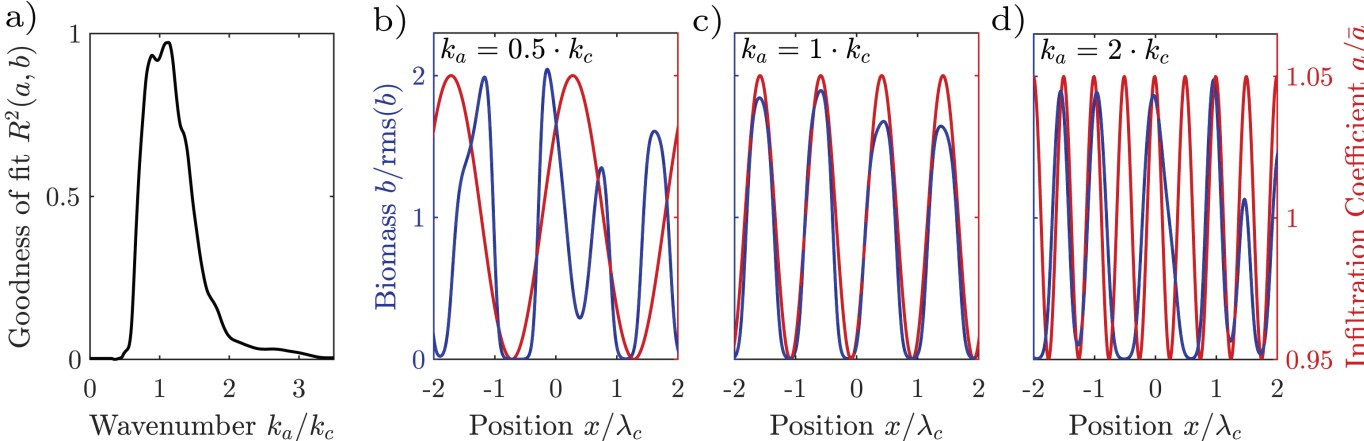

**Fig 8. Frequency response of the Rietkerk model.** (a) Correlation of a one-dimensional pattern generated with the Rietkerk model, where the infiltration coefficient $a$ varies sinusoidally with a varying frequency. The coefficient $a$ varies sinusoidally in space with wavenumber $k_a = 2\pi/\lambda_a$, a mean $\bar{a}$ of 0.2 mm/d and a relative amplitude of 5%. The correlation is highest, when the coefficient oscillates at a wavelength close to that of the pattern $\lambda_a \approx \lambda_c$. (b–d) Parts of the pattern generated by varying the infiltration coefficient sinusoidally. In case the wavenumber of the heterogeneity matches that of the pattern, the vegetation stripes align with the infiltration coefficient ($d_{ii}$), and the pattern is highly correlated with the infiltration coefficient ($a$), so that the vegetation pattern is imposed by the exogenous heterogeneity. The vegetation stripes do not align with the infiltration coefficient and the correlation between the pattern and the infiltration coefficient is low when the infiltration coefficient varies at a wavelength considerably smaller ($d_i$) or larger ($d_{iii}$) than the wavelength of the pattern.

The spectral coherence indicates that the pattern formation can be reproduced by a linear filter to a large degree. Averaged over the spectrum, the spectral coherence is larger 0.5 when the exogenous hetoregeneity is not too small $cv(a) \geq 0.03$. The spectral coherence is largest at intermediate degrees of heterogeneity, with a value of 0.66, 0.75 for labyrinthine and 0.77 for gapped patterns, Fig 7c. The spectral coherence decreases towards higher degrees of heterogeneity slightly. For all patterns, the spectral coherence is lower for components in the low frequency tail ($k_r < k_c$) then for components in the high frequency tail ($k_r > k_c$), Fig 7d. The transfer function determined through the cross-spectral density Eq 13a had an imaginary part close to zero for all isotropic patterns generated with the Rietkerk model, confirming that it is reasonable to assume a zero phase shift when estimating a linear filter in cases where the spatial map of the exogenous heterogeneity is not known. Consistent with the spectral coherence, the squared transfer $|T|^2$ function has a similar shape as the spectral density $\mathcal{S}$, but a smaller magnitude than the spectral density in the high frequency tail, indicating that nonlinear interaction predominantly generate high frequency components which are associated with the sharp transition between vegetation patches and bare ground. Motivated by the similar behaviour of the Rietkerk model to a bandpass filters and the high spectral coherence between the exogenous heterogeneity and the pattern, we derive a parametric linear bandpass filter.

## Linear band-pass filter

A linear filter can be found for the Rietkerk model by linearization, c.f. section . However, the expression of the linear filter is extensive and contains many parameters. We therefore propose a simple band-pass filter, which captures the main characteristics of the response of the Rietkerk model to exogenous heterogeneities. We again report only the results here, and provide its derivation in the supplement section 2.3. We therefore approximate the response with a simple band-pass filter with two parameters, which amplifies spatial components close to the characteristic wavelength, and attenuates spatial components with shorter and longer wavelengths. The bandpass filter can generate patterns with a wide range of regularity which appear similar to natural patterns, Fig 9a$_i$–Fig 9a$_{iii}$. The spectral density of the band-pass filter has a single lobe with a maximum at the characteristic wavenumber $k_c$, Fig 9b. The term band-pass derives from this lobe, which is also referred to as a frequency band, and should not be confused with stripes of anisotropic patterns. The two parameters of the filter determine its characteristic wavelength $\lambda_c$ and the regularity $S_{rc}/\lambda_c$. The band-pass filter can therefore generate patterns over a wide range of regularities, Fig 9a. The spectral density of the band-pass filter fits well the densities of natural patterns, $R^2 = 0.98$, Fig 9d. It also fits well the density of patterns generated with the Rietkerk model irrespectively of the pattern type or the degrees of exogenous heterogeneity, $R^2 > 0.89$, Fig 8c.

The band-pass filter does not only reproduce spatial correlation structure of a pattern, but can even predict the location of individual patches. We demonstrate this by generating again pairs of patterns, one with Rietkerk model, and one by band-pass filtering. We first fit the band-pass parameters to the pattern generated with the Rietkerk model, and then filter the same spatial map of the noisy coefficient, and then compare the thresholded patterns. The patterns generated with both methods look very similar and overlap to a large degree, Fig 9d. The patches are located in both patterns at similar locations, and only slightly differ in size and shape. The patterns are also strongly correlated. The bandpass filter predicts the patterns of all types well, when the degree of heterogeneity is sufficiently large ($R^2 \leq 0.85$ for $cv(a) \leq 0.1$).

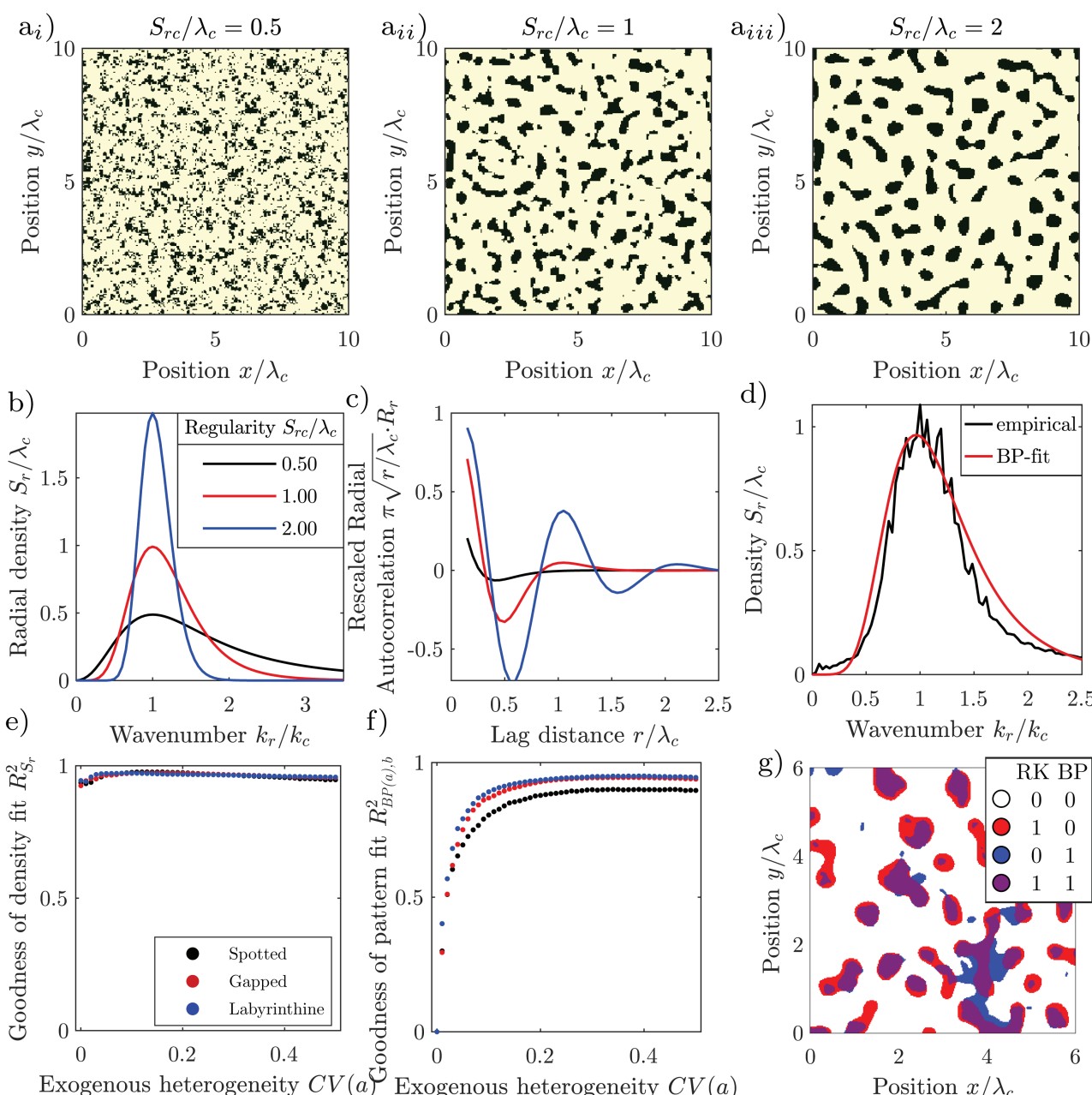

**Fig 9. Generic isotropic patterns.** ($a_i$–$a_{iii}$) Patterns with several degrees of regularity $S_{rc}/\lambda_c$ generated by band-pass filtering a spatially varying heterogeneity and subsequent thresholding. (b) Corresponding radial spectral densities and (c) autocorrelation. The autocorrelation is corrected for the spurious decay introduced by radial averaging. (d) Fit of the bandpass spectral density to the radial density of the natural regular isotropic pattern in Fig 1b$_i$. (e) Goodness of fit of the bandpass spectral density the radial spectral density of the isotropic regular patterns generated with the Rietkerk model, depending on the degree of exogenous heterogeneity cv($a$). Parameter settings identical to those in Fig. 7. (f) Goodness of fit between patterns generated by bandpass filtering and patterns generated with the Rietkerk model for the same spatial heterogeneity map. (g) Overlap and difference thresholded spotted patterns generated with the Rietkerk model (RK) and generated by band-pass (BP) filtering the spatially infiltration coefficient. The spatial structure of both patterns agrees well (cv($a$) = 0.3).

Reaction-diffusion models, with scale-dependent feedbacks, such as the Rietkerk model, therefore act like band-pass filters in spatially heterogeneous environments, generating patterns by amplifying and suppressing spatial components of the heterogeneity depending on their wavelength. A pattern can therefore be reliably predicted by a linear filter and subsequent thresholding, as long as the parameters, i.e., the characteristic wavelength, regularity and fraction of the ground covered by vegetation are known. The main influence of the nonlinearity during the pattern formation is that it determines the filter parameters, while the spectral density and hence spatial structure of the filter are always similar to that of a linear bandpass filter.

## Anisotropic regular patterns

Anisotropic, i.e., striped, regular patterns form at locations where advection, i.e., unidirectional flow is relevant [6,17,78,79]. At semi-arid hillslopes, for example, the flow of water causes the vegetation stripes to grow parallel to the elevation contours [80]. The stripes slowly migrate uphill over time, with new vegetation establishing at the uphill front of stripes which receives runoff from the bare uphill area, and old vegetation dying back at the dryer downhill tail of the stripes [65]. We illustrate this based on a semi-arid vegetation pattern from Kordofan, Sudan, Fig 1c$_i$. The width of vegetation stripes, the distance between them and their direction varies. Their fringes are ragged. Stripes occasionally terminate or split in two. The frequency components are scattered in a region surrounding the central wavenumber, Fig 1c$_{ii}$. The autocorrelation oscillates with decaying amplitude, indicating that the pattern decorrelates with increasing lag distance, Fig 1c$_{iii}$. The regularity is intermediate ($S_{x^+c}/\lambda_c$ = 1.3, $S_{y^+c}/\lambda_c$ = 3.4) and the statistical test does not classify the pattern as periodic ($p$ = 0.15).

The Rietkerk model generates regular striped patterns when the advection of water is accounted for. When the environment is homogeneous, the pattern is highly regular consisting of stripes that are equally wide, spaced in equal intervals, have smooth fringes, run in parallel and neither split nor terminate, Fig 10a$_i$. Most of the spectral density consists of two narrow peaks with wavenumber $k_x = \pm k_c$, Fig 10a$_{ii}$. The pattern is spatially correlated over its entire extent, Fig 10a$_{iii}$. The regularity is very high ($S_{x^+c}/\lambda_c$ = 11.6, $S_{y^+c}/\lambda_c$ = 13.4) and the statistical test classifies the pattern as periodic $p$ = 0.00. The pattern generated by the Rietkerk model in a homogeneous environment is therefore considerably more regular than the natural pattern. The model does neither reproduce the variation of the stripe width and spacing, nor the scatter of the frequency components, nor the decay of the spatial autocorrelation.

## Influence of exogenous heterogeneity on the formation of anisotropic regular patterns

To reproduce the spatial structure of the natural striped pattern, we again introduce spatial heterogeneity by randomly varying the infiltration coefficient $a$. The Rietkerk model indeed generates patterns which are more similar to natural ones when the coefficient is spatially varying, Fig 10b$_i$. For an intermediate degree of regularity (cv($a$) = 0.1), the width and distance between stripes varies, and they occasionally terminate and split. The frequency components are scattered in two lobes centred at $\pm k_c$, Fig 10b$_{ii}$. The correlogram oscillates, but the oscillation decays with increasing lag distance within a few wavelengths, Fig 10b$_i$. The regularity is intermediate ($S_{x^+c}/\lambda_c$ = 1.5, $S_{y^+c}/\lambda_c$ = 1.3) and the statistical test does not classify the pattern as periodic ($p$ = 0.08).

We systematically explore how the Rietkerk model responds to spatial noise, by generating a series of patterns with different degrees of spatial heterogeneity cv($a$) of the infiltration coefficient $a$, Fig 12a$_i$. Without noise cv($a$) = 0, the pattern is highly regular

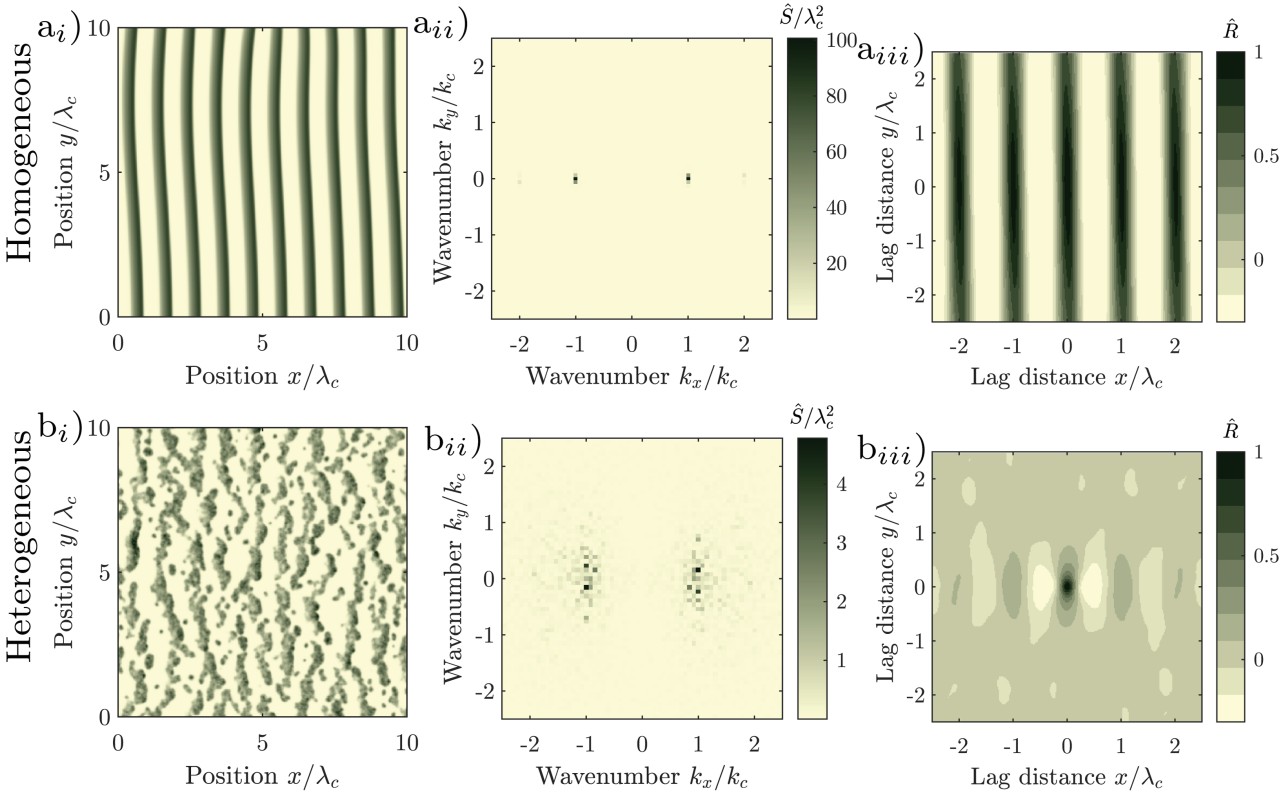

**Fig 10. Anisotropic regular patterns generated with the Rietkerk model (rainfall intensity $R = 1$ mm/d, runoff velocity $v_x = 10$ m/day, and water diffusion $e_x = 0$, $e_y = 20$ m$^2$/day, other parameters are set to the same values specified in Fig 6).** ($a_i$) A highly regular pattern generated in a homogeneous environment (cv($a$) = 0), regularity $S_{x^+c}/\lambda_c = 11.6$, $S_{y^+c}/\lambda_c = 13.4$, $p$-value of periodicity test = 0.00. ($b_i$) a less regular pattern with spatially varying infiltration coefficient (cv($a$) = 0.1), regularity $S_{x^+c}/\lambda_c = 1.52$, $S_{y^+c}/\lambda_c = 1.3$, $p$-value of periodicity test = 0.08. ($a_{ii}$, $b_{ii}$) corresponding periodograms, ($a_{iii}$, $b_{iii}$) corresponding correlograms.

($S_{x^+c}/\lambda_c = 11.6$), with a magnitude close to the theoretical maximum of $L/\lambda_c$, where $L$ is the side length of the modelled domain. The pattern is also classified as periodic by the statistical test. The characteristic wavelength is 73 m. With increasing exogenous heterogeneity ($0 < $ cv($a$) $ < 0.1$), the regularity drops rapidly, reaching $S_{x^+c}/\lambda_c = 0.42$ at cv($a$) = 0.15. The regularity decreases due to a rapid decrease of the maximum $S_c$, only slightly influenced by the wavelength which increases slightly to 100 m over the same range.

The statistical test classifies the patterns as not periodic when the degree of exogenous heterogeneity exceeds cv($a$) = 0.06. At this point the exogenous heterogeneity only causes 10% to the total heterogeneity of the infiltration, while most of it is caused by the vegetation feedback, Fig 11a. For intermediate degrees of exogenous heterogeneities ($0.15 < $ cv($a$) $ < 0.42$), the regularity and wavelength level out and stay close to $S_{x^+c}/\lambda_c = 0.4$ and $\lambda_c = 100$ m. Beyond cv($a$) = 0.42 the regularity drops to zero due to a rapid increase of the wavelength. For relatively high degrees of exogenous heterogeneity, vegetation stripes break into smaller patches and cease to migrate uphill, Fig 11b. Under such conditions patterns appear more similar to isotropic patterns. Over all, the influence of the exogenous heterogeneity is similar for anisotropic and isotropic patterns, the regularity decreases and the wavelength increases. However, for isotropic patterns the effect of the exogenous heterogeneity is strong already for small degrees of heterogeneity, while for anisotropic patterns the effect of the exogenous heterogeneity is

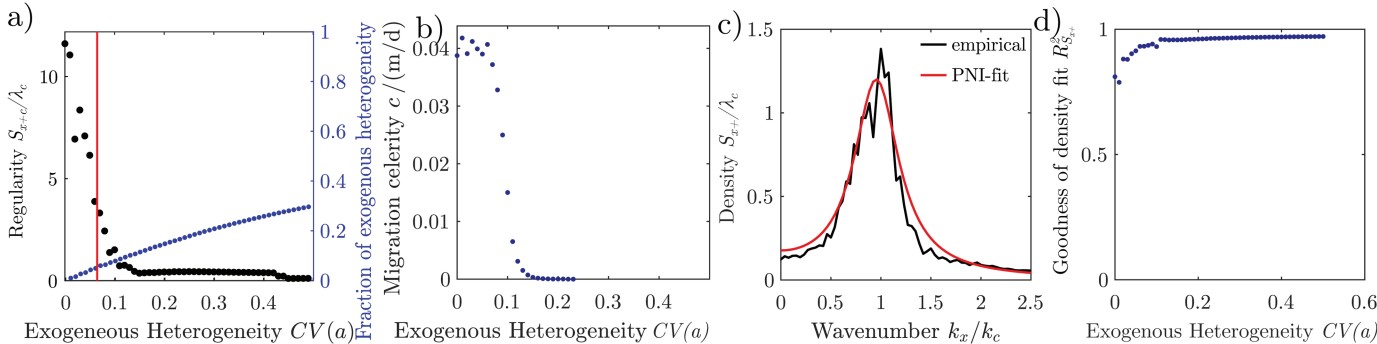

**Fig 11. Response of the Rietkerk model to environmental heterogeneity at hillslopes.** (**a**) Regularity of anisotropic patterns generated with the Rietkerk model and (**b**) the migration celerity of vegetation stripes depending on the degree of random exogenous spatial heterogeneity of the infiltration coefficient cv($a$). Model parameters set to the same values as in Fig 10. The red vertical line in a) indicates the amount of heterogeneity beyond which patterns are not any more periodic at a 0.05 confidence level. (**c**) Fit of the noisy oscillator density to the density perpendicular to the stripes of the natural anisotropic pattern in Fig 1c$_{ii}$. d) Goodness of fit of the oscillator density to the density perpendicular to the stripes of the anisotropic patterns generated with the Rietkerk model, c.f. Fig 12.

somewhat less strong for small degrees of heterogeneity but then abruptly intensifying around cv($a$) ≈ 0.1, which is likely related of the transition from a migrating to a stationary pattern.

## Anisotropic regular pattern formation as a noisy oscillation

The influence of random exogenous spatial heterogeneities on the formation of anisotropic patterns can be understood by viewing the system as an oscillator. In the case of dryland ecosystems, vegetation stripes are spaced at similar intervals, resulting in a cyclic, i.e., oscillatory, alternation between bare ground and vegetation. This is because a vegetation stripe requires an uphill bare area for catching a sufficient amount of precipitation for sustaining itself [81]. The distance between stripes and the width of stripes is similar throughout the pattern, as this optimizes the water usage. When one vegetation stripe is displaced, then the next stripe further downhill shifts accordingly to keep the length of the bare ground between the stripes optimal, and subsequently all stripes further downhill shift one after the other. This is similar to a clock: An unperturbed clock runs evenly. When it is perturbed, it continues to run evenly. However, it does not correct the incurred shift. We demonstrate this behaviour for striped vegetation patterns with a numerical experiment with the Rietkerk model in one dimension. We reduce the infiltration coefficient at a single location. At the perturbation, the wavelength, amplitude and phase locally deviate from that further uphill, Fig 12a$_{ii}$. The wavelength measures the interval at which vegetation stripes occur, the amplitude measures the maximum vegetation density in the stripes, and the phase indicates the position of the stripes. Further downhill, the amplitude and wavelength of the pattern return to the uphill values as if no perturbation had occurred. However, the phase does not recover. The stripes downhill of the perturbation remain shifted with respect to the position they would have were there no perturbation, Fig 12a$_i$. The amplitude and wavelength recover because they are constrained by the environmental conditions. However, the phase of a pattern is not constrained, as it cyclically changes over time as the stripes migrate uphill.

When a pattern is perturbed in several places, then the displacements, i.e., the phase error, integrates along the hillslope, and the pattern decorrelates with increasing distance. When the pattern is randomly perturbed at every point along a transect, then the phase resembles a random walk with drift, where the drift corresponds to the deterministic change of the phase driven by the scale-dependent feedbacks which align stripes periodically in homogeneous

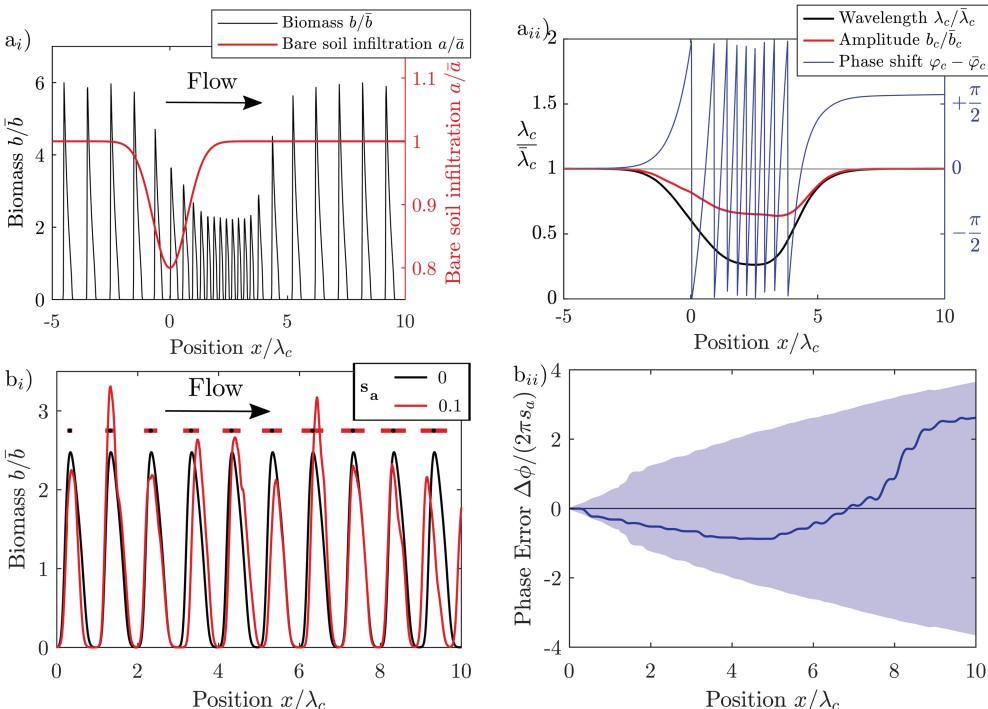

**Fig 12. Similarity of striped vegetation patterns with noisy oscillators.** ($a_i$) Influence of a localized perturbation of the infiltration coefficients on a striped pattern. ($a_{ii}$) corresponding local amplitude, phase shift and wavelength. The wavelength and amplitude are locally reduced where the infiltration is inhibited but recover further downhill. In contrast, the phase of the pattern does not recover. Consecutive perturbations in a heterogeneous environment thus accumulate along the pattern and result in a random walk of its phase. ($b_i$) Influence of a randomly perturbed infiltration coefficient ($cv(a) = 0.1$), precipitation $R = 1$ mm/d, surface water advection $v_x = 10$ m/d, surface water diffusion $e_x = 0$. Horizontal bars indicated $\pm 1$ standard deviation of the expected phase deviation. ($b_{ii}$) Corresponding phase deviation of the same pattern with respect to a pattern forming in an idealized homogeneous environment. The shaded area indicates the standard deviation of the expected phase deviation, determined from 100 model runs.

environments, and the random walk originating from the integrated random displacements. This is similar to phase noise integration in oscillators which is an important phenomenon in the field of electronic engineering, where random perturbations result in the decorrelation of the oscillation over time [82–84].

To demonstrate this, we conduct another numerical experiment with the Rietkerk model in one dimension, where randomly vary the infiltration coefficient in one half of the spatial domain. In the half with constant coefficient, a nearly periodic sequence of stripes forms which over time migrate into the half of the domain with heterogeneous infiltration. In the half of the domain with varying infiltration, the amplitude, width and distance between the vegetation stripes varies, Fig 12$b_i$. In particular the phase $\phi$, i.e., the position of the stripes, deviates from that of a periodic pattern as it undergoes a random walk with drift. With respect to the phase $\phi_0$ at the upstream end of the heterogeneous section, $\Delta\phi = \phi - \phi_0 - k_c x$, where $\Delta\phi$ is the deviation of the phase with respect to the deterministic drift $k_c x$ of the phase of a periodic pattern. When the degree of variation of the infiltration coefficient is small, then the random walk roughly follows the along-slope integral of the infiltration coefficient, i.e., $\Delta\phi \propto \int (a(x) - \bar{a})\,dx$, where $\bar{a}$ is the mean of the infiltration coefficient and has the variance $E[(\Delta\phi)^2] \propto (x - x_0)^2$, which is also the case in our experiment with the Rietkerk model 12$b_{ii}$.

Phase noise integration is likely relevant to other spatial systems where stripes are oriented perpendicular to the flow, such as mussel beds and seagrass [22]. In some systems stripes are aligned perpendicular to the flow, for example patterns consisting of fluvial macrophytes [78] and algae [79]. In such systems regularity could potentially be determined by band-pass filtering in the direction perpendicular to stripes and low-pass filtering in the direction parallel to stripes, rather than phase noise integration.

Phase noise integration explains why random exogenous spatial heterogeneities change the spatial structure of regular anisotropic patterns. However, other factors, such as variation of the amplitude and wavelength also play a role. Phase noise integration therefore does not predict the location of stripes in a particular pattern well. Still we will use the phase noise integration as a model to find linear filters which generate striped patterns with similar spatial structure as that of natural patterns and patterns generated with nonlinear models.

### An oscillator model with phase noise

A linear filter generating anisotropic patterns can be found by directly linearizing the Rietkerk model. However, we again derive a simpler model, as the expression of the linearization yields complicated expressions. A different model than band-pass filtering is required, as the spectral density of anisotropic patterns is different to that of isotropic patterns, as it has heavier tails. In particular, the spectral density of anisotropic patterns remains positive at the origin ($\lim_{k_x \to 0} S_x(k_x) > 0$), while the spectral density of isotropic patterns drops to zero ($\lim_{k_r \to 0} S_r(k_r) = 0$). Furthermore, the spectral density between the two axes, perpendicular and parallel to the stripes, has to be distinguished. As before, we derive a parsimonious linear filter, a filter with just three parameters determining the characteristic wavelength, as well as the regularity in the direction perpendicular and parallel to the stripes. We again report only the results here, and provide the detailed equations in the supplement, Section 2.4.

We model anisotropic pattern formation as a nonlinear oscillation with constant wavelength and amplitude where the phase deviates due to the integration of noise. The shift of the position of stripes with respect to an unperturbed pattern, is described by a random walk, i.e., Brownian motion in one dimension, and a Brownian surface in two dimensions. The simple oscillator has a drift term for the systematic component of the phase change in the direction perpendicular to the bands. The patterns generated by the nonlinear oscillator have a bimodal biomass distribution and thus do not have to be thresholded as patterns generated by the linear filters. The patterns cannot be directly generated by convolving noise with the autocorrelation or correspondingly multiplying noise with the transfer function in the frequency domain. Instead, patterns can be generated by determining the phase deviation first, which can be facilitated with Stein's method [85]. The spectral density of the nonlinear oscillator has a simple parametric expression, which yields a linear filter. The linear filter can be used to generate patterns with similar spatial structure as that of the nonlinear oscillator, without having to determine the phase error explicitly, but the pattern has to be thresholded as the patterns generated by the other linear filters.

The nonlinear oscillator has three parameters, determining the length scale $\lambda_c$, as well as the regularity in the directions perpendicular $S_{x+c}/\lambda_c$ and parallel $S_{y+c}/\lambda_c$ to the stripes. The oscillator can therefore reproduce anisotropic regularities. The spectral density of the oscillator consists of a single lobe in the direction perpendicular to the stripes, Fig 13e and appears similar to that of the band-pass, Fig 9. The spectral density of the nonlinear oscillator has heavier tails than the spectral density of the bandpass filter, i.e., more of the spatial variance is contributed by components with wavenumbers that are much lower of much higher than the characteristic wavenumber.

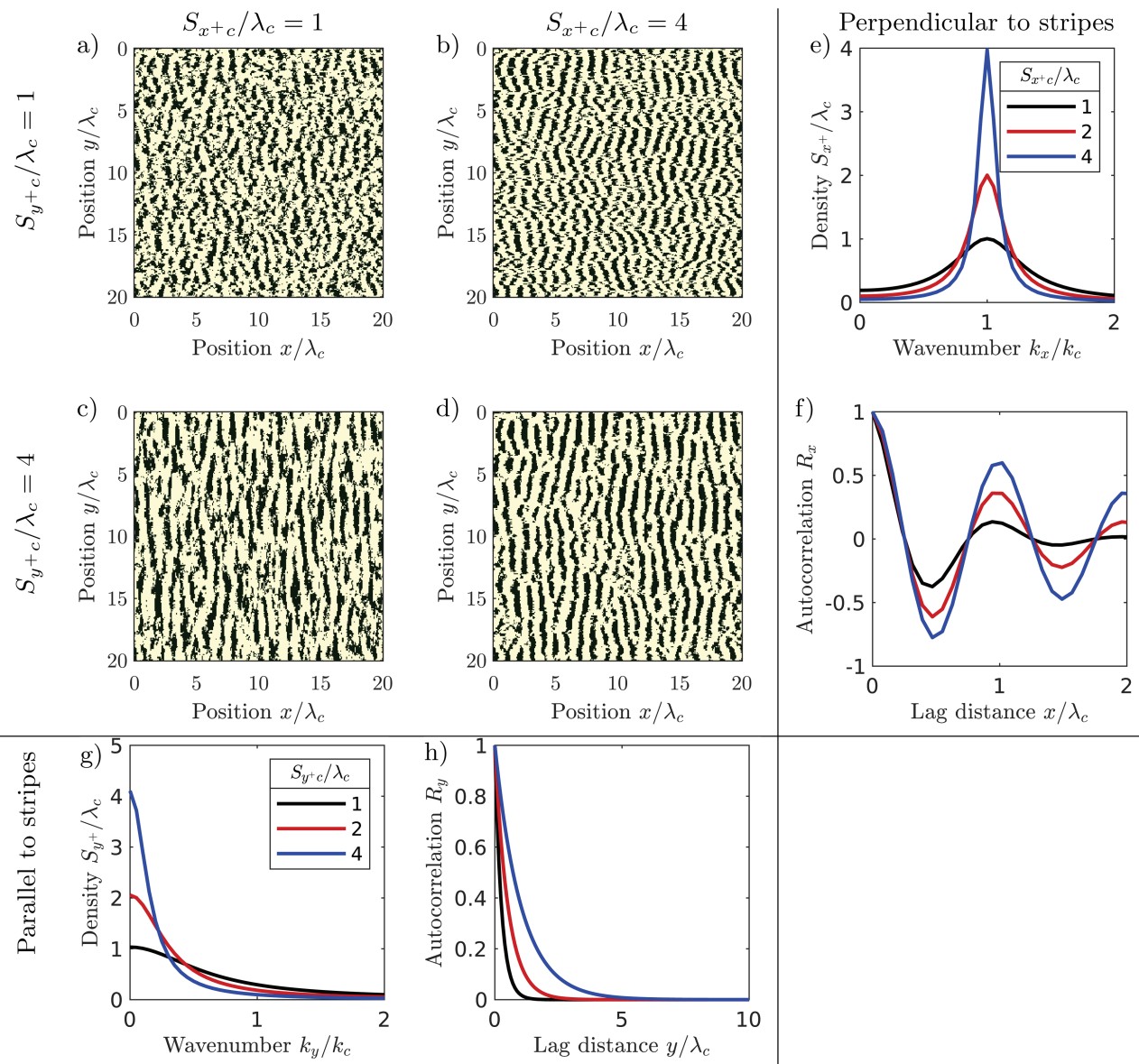

**Fig 13. Generic anisotropic patterns.** (a–d) Striped patterns generated with the nonlinear oscillator for varying degrees of regularity. (e–f) Corresponding densities $S_x$ and autocorrelation $R_x$ in the direction perpendicular to the stripes. (g–h) Corresponding densities $S_y$ and autocorrelation $R_y$ in the direction parallel to the stripes. The two-dimensional density and autocorrelation are given by the outer product of the one-dimensional densities. The more regular a pattern, the higher the maximum $S_{xyc} \approx S_{x+c} S_{y+c}/(4\lambda_c^2)$ of the spectral density, and the less rapidly the autocorrelation $R_{xy} = R_x \otimes R_y$ decays. The regularity of striped patterns can be anisotropic, i.e., it differs between the directions parallel and perpendicular to the stripes. Therefore, two parameters are necessary to unambiguously specify the regularity. This leads to a larger variety of striped patterns compared to isotropic spot and gap patterns, where the regularity is uniquely determined by a single parameter.

In particular, the spectral density does not go to zero at the origin when the regularity is intermediate ($S_x(0) > 0$). The heavier tails cause the fringes of vegetation stripes to be more ragged. The spectral density in the direction parallel to the stripes has its maximum at the origin and decays towards higher wavenumbers, Fig 13g. A patterns is the more regular, the larger the product $S_{x+c} \cdot S_{y+c}/\lambda_c^2$, Fig 13a,d. When the anisotropy $S_{x+c}/S_{y+c}$ of the regularity is large, then the edges of the stripes are more ragged, but the distance between stripes is more

regular, Fig 13b. When the anisotropy $S_{x^+c}/S_{y^+c}$ of the regularity is small, then the edges of the stripes are smoother, but the distance between the stripes varies more strongly, Fig 13c.

The patterns generated by the nonlinear oscillator, Fig 13a–d, appear visually similar to natural patterns, Fig 1c$_i$, and patterns generated with the Rietkerk model in heterogeneous environments, Fig 13a–d. The spectral density of the nonlinear oscillator fits the spectral density of natural patterns, $R^2 = 0.97$, Fig 11c. It also fits the density of patterns generated with the Rietkerk model well, Fig 11d, with $R^2 = 0.96$ for patterns of intermediate and low regularity and $R^2 = 0.81$ for highly regular patterns.

## Discussion

Regular environmental spatial patterns can form through self-organization in systems with scale dependent biophysical feedbacks [4,6,25,27]. In idealized homogeneous environments patterns form autogenously, which has been demonstrated with deterministic reaction-advection-diffusion models, in analogy to the morphogenesis of complex organisms [1]. As patterns can form autogenously in homogeneous environments, modelling studies of environmental spatial patterns usually neglect exogenous heterogeneity. However, these autogenously forming patterns are considerably more regular than natural patterns, and hence do not reproduce their spatial structure well [34,35]. This is because exogenous heterogeneity, such as the variation of soil spatial properties [41,51,86], or microtopography [42], considerably influences the formation of environmental spatial patterns. This differs from the morphogenesis of individual organisms, where perturbations only excite the initial pattern formation. For organisms, the surface-to-volume ratio and with it, the magnitude of exogenous perturbations relative to their size decreases exponentially over time. This is not the case for environmental spatial patterns, as they have a nearly two-dimensional spatial extent. Both endogenous and exogenous factors consequently affect how environmental spatial patterns form. Large-scale topographic heterogeneities can considerably influence the formation of spatial patterns, in particular when they limit the spatial extent of a pattern [87–90].

The influence of small random spatial heterogeneities is less obvious [42,51,52,54,55,87,89, 91]. The main effect of random exogenous heterogeneity is that it makes patterns less regular, i.e., it widens the spectral density, and decorrelates the pattern with increasing distance. The regularity of natural patterns is thus in between the extreme limits of periodic patterns on one end, where the spatial variance is concentrated in a narrow and high peak peak that is well separated from the origin, and irregular patterns at the other end, where the spatial variance is concentrated over a wide range of frequency components around the origin.

### Regular pattern formation through filtering of noise

Natural processes are inherently noisy [10,92]. In many cases, the influence of the noise is negligible so that the processes can be understood from a deterministic perspective. In contrast, the formation of environmental spatial patterns is highly sensitive to noise. Regular patterns can thus form through stochastic processes similar to irregular patterns [5,57,73]. The spatial structure of natural spatial patterns can consequently only be well reproduced by a model when it accounts for exogenous random spatial heterogeneity. Our study shows that isotropic patterns can form through filtering of random exogenous spatial heterogeneities. Self-organization in the sense is the suppression and amplification of frequency components of exogenous spatial heterogeneity, depending on the spectral density of the pattern forming system. This increases order and decreases the entropy, and in the case of regular patterns, increases regularity. The difference between regular and irregular patterns is that irregular patterns form through the suppression of low frequency components, while regular patterns

form through the suppression of both low and high frequency components, c.f. Fig 14. Components of the exogenous heterogeneity with a wavelength close to the characteristic wavelength of the system therefore serve as a template for emerging patterns. In this sense irregular patterns are no less self-organizing than regular patterns.

While a nonlinear system responds differently to a single perturbation depending on its state at the point of perturbation, it responds to small random perturbations at many locations statistically in a similar manner, i.e., when a pattern is split into tiles, then the spectral density, and hence the spatial structure off all tiles will be similar, as long as the tiles are sufficiently larger than the patch size. For every regular isotropic pattern a stable linear filter can

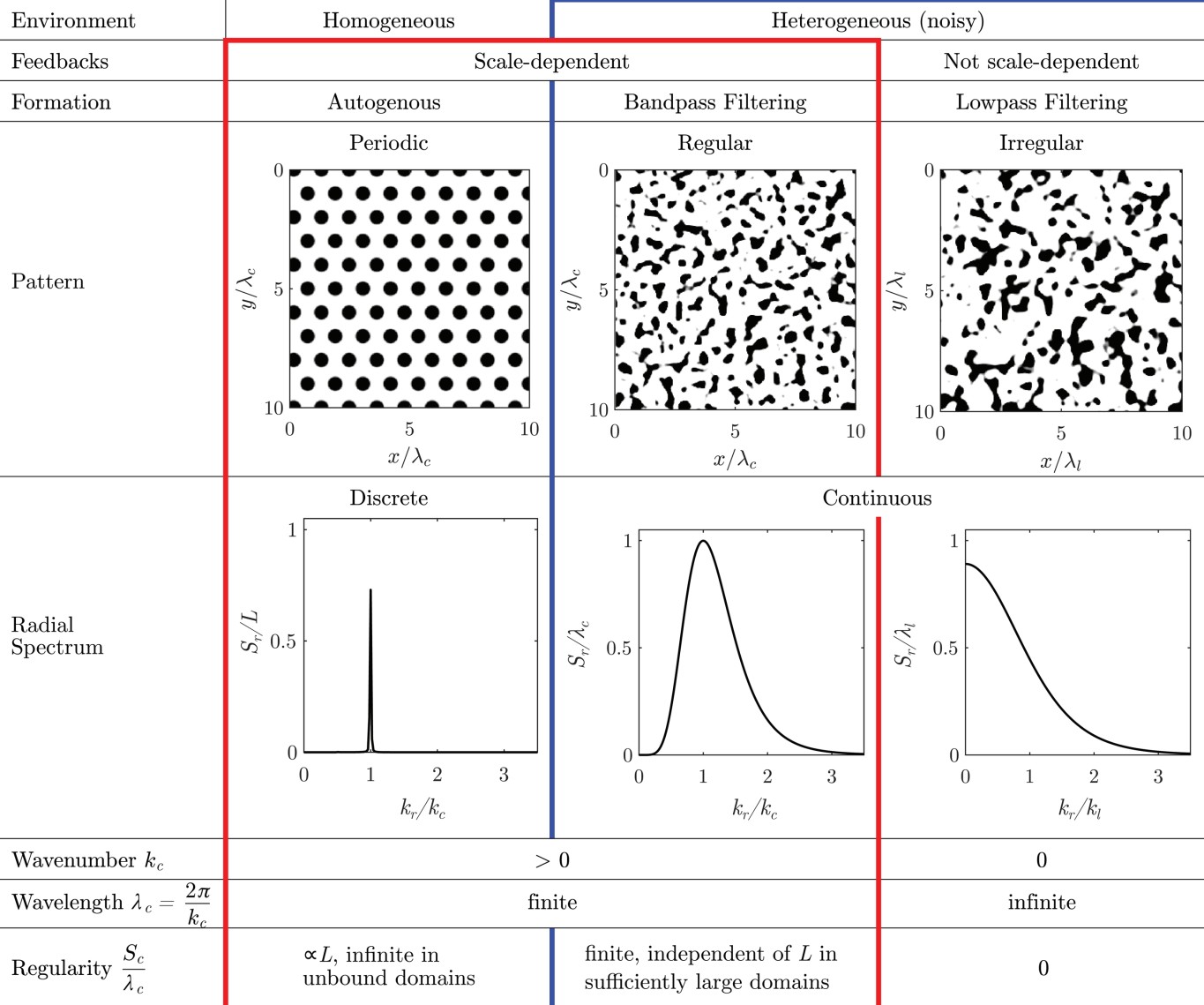

**Fig 14. Formation of spatial patterns under various conditions**: Patterns forming in homogeneous environments through scale-dependent feedbacks become periodic. Patterns forming in heterogeneous environments through scale-dependent feedbacks become regular, but not periodic. Patterns forming in heterogeneous environments without scale-dependent feedbacks become irregular.

be found by approximating the transfer function of the filter as the square root of the pattern's spectral density. This filter generates patterns with similar spatial correlation structure as the prototype. For the Rietkerk model, the frequency spectra of patterns is largely coherent with the spectra of the spatially varying infiltration coefficient, which indicates that the patterns can be well predicted by linear filters. This is remarkable as the response of nonlinear systems to noise in many other cases cannot be well reproduced by linear approximations [93], in particular not for anisotropic vegetation patterns.

Linearization might seem a drastic simplification, in particular as the conditions for pattern formation can only be understood as a nonlinear process. However, it should be kept in mind that most nonlinear models also extremely simplify the complex biophysical processes. They usually consider only a few state variables and express their interactions through simple multivariate polynomials [6,75]. In consequence, linear filters can better reproduce the spatial structure of natural regular spatial patterns compared to simple nonlinear models assuming idealized homogeneous environments [35]. Nonlinearities mostly contribute to high frequency components which are associated with the sharp transition between vegetation patches and bare ground. This can be mimicked by thresholding a pattern predicted by a linear filter.

A parametric linear filter can be found by directly linearizing a nonlinear system, but the resulting expressions are cumbersome, and not insightful, as the linearization at many points has to be statistically averaged. We therefore propose a simple linear bandpass filter based on the conceptual behaviour of isotropic pattern forming systems. The linear filter reproduces the main spatial characteristics of the patterns, their characteristic wavelength and regularity. The parameters of the linear filter can be estimated from an orthographic image of a spatial pattern. We find that a linear bandpass filter can predict the location of individual patches of isotropic patterns generated by the Rietkerk model reasonably well. The linear filter does predict the phase of the frequency components of a pattern, however, the phase of the linearly predicted components is identical to that of the corresponding components of the exogenous heterogeneity, and thus random.

While linear filters cannot predict the bimodality of the statistical biomass distribution, they still can predict the spatial structure of patterns. This can be understood by recalling that the correlation structure neither depends on mean nor the variance of the statistical biomass distribution, nor on the phase of the frequency components. Differences between vegetation patterns generated with linear filters and nonlinear models are small, but distinct: patches, gaps and rings generated by linear filters appear slightly less round, and the cross-section of vegetation stripes is not asymmetric. Though, these characteristics are not strongly pronounced in most natural patterns, as they have an intermediate degree of regularity. Linearization also does not reproduce the periodic hexagonal grid structure of isotropic patterns which crystallizes in idealized homogeneous environments. However, such conditions are rarely found in nature. The patterns generated by linear filters have furthermore six neighbours on average, as this an intrinsic property of any random point pattern in two dimensions. Only the direction in which neighbours are located varies throughout the pattern. Lastly, while linear filters can reproduce the spatial structure of patterns, the parameters of the filters are nonlinear functions of the parameters of the nonlinear system. However, the parameters of the linear system can be estimated from a natural pattern without that the nonlinearity has to be modelled by itself.

Anisotropic systems resemble oscillators where the phase undergoes a random walk under the influence of exogenous heterogeneity. The location of migrating vegetation stripes cannot be directly predicted from the heterogeneity. The location of stripes is only predictable when the stripes remain stationary, which is the case when the magnitude of the heterogeneity is

very large. However, we still provide a simple oscillator model which can generate anisotropic patterns with similar spatial structure as a prototype. A particular pattern cannot be predicted as the problem is not well posed. This is not limited to the prediction by linear models. The anisotropic Rietkerk model will generate entirely different patterns when the flow velocity is slightly perturbed. In contrast, the prediction of isotropic patterns is a well posed problem and the Rietkerk model will converge to nearly the same state when the diffusion coefficient is slightly perturbed.

## Relevance of exogenous heterogeneities for pattern forming ecosystems

Exogenous random heterogeneities influence the spatial structure of environmental spatial patterns, but could also be relevant for the function and resilience of pattern forming ecosystems. Environmental heterogeneity could potentially increase the resilience of ecosystems, as they decrease regularity, i.e., the distance over which patterns are spatially correlated, thereby making sudden catastrophic shifts of entire ecosystems less likely. Semi-arid vegetation is conventionally perceived to have a regular spatial structure [94], and ecosystem functioning and resilience is predominantly studied with models that generate highly regular patterns, i.e., models that emulate homogeneous environments [16,95]. The spatial structure of patterns has been suggested as an indicator of desertification [96], as water scarcity increases the competition between vegetation patches, strengthening the pattern-forming feedback [71]. In heterogeneous environments, the spatial structure of patterns does not strongly depend on the water availability, as it can be generically predicted by linear filters, independently of the fraction of the ground covered by vegetation. Both regular and irregular patterns are furthermore abundant in drylands, sometimes even in close proximity. Differences of pattern regularity between two locations thus might just be caused by different degrees of spatial heterogeneity, and not necessarily by differences in water scarcity. For example, due to differences between rugged hillslopes consisting of leptosols on one hand and flat valley floors consisting of alluvium. The spatial structure of highly regular patterns has also been suggested as an indicator of past climate, as it is history dependent [28,39,97–99]. We noticed that this dependence is drastically reduced in simulations with random exogenous heterogeneities. The spatial structure might therefore be less informative about the health and history of ecosystems than commonly assumed. Asserting how exogenous random heterogeneities influence the functioning and resilience of pattern forming ecosystems requires further research: What are the sources of exogenous heterogeneity. What is their magnitude and spatio-temporal structure? Which spatial and temporal scales have to be resolved? Which processes have to be considered? Does the influence of vegetation on the microtopography and soil properties matter?

## Supporting information

**S1 Appendix.** Detailed derivation and analysis of filters.
(PDF)

## Author contributions

**Conceptualization:** Karl Kästner.

**Formal analysis:** Karl Kästner.

**Investigation:** Karl Kästner.

**Methodology:** Karl Kästner.

**Software:** Karl Kästner.

**Writing – original draft:** Karl Kästner.

**Writing – review & editing:** Karl Kästner, Daniel Caviedes-Voullième, Christoph Hinz.

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
