## [Decision Letter · Decision Letter 0]

29 Nov 2024

PONE-D-24-45670Spatial pattern formation in heterogeneous environmentsPLOS ONE

Dear Dr. Kästner,

Thank you for submitting your manuscript to PLOS ONE. After careful consideration, we feel that it has merit but does not fully meet PLOS ONE’s publication criteria as it currently stands. Therefore, we invite you to submit a revised version of the manuscript that addresses the points raised during the review process.

We look forward to receiving your revised manuscript.

Kind regards,

Pan Li, PhD

Academic Editor

PLOS ONE

Journal Requirements:

Reviewers' comments:

Reviewer's Responses to Questions

**Comments to the Author**

1. Is the manuscript technically sound, and do the data support the conclusions?

Reviewer #1: Partly

Reviewer #2: Yes

Reviewer #3: Yes

2. Has the statistical analysis been performed appropriately and rigorously? 

Reviewer #1: I Don't Know

Reviewer #2: Yes

Reviewer #3: N/A

3. Have the authors made all data underlying the findings in their manuscript fully available?

Reviewer #1: Yes

Reviewer #2: Yes

Reviewer #3: Yes

4. Is the manuscript presented in an intelligible fashion and written in standard English?

Reviewer #1: Yes

Reviewer #2: No

Reviewer #3: Yes

5. Review Comments to the Author

Reviewer #1: The authors explore the formation of irregular and regular (but not perfect) vegetation patterns, suggesting that their formation resembles the actuation of a filter on top of heterogeneities.

The manuscript is not easy to follow, and the relevance of the findings is not clear to me. The authors claim that vegetation patterns in nature are not perfectly periodic as the ones formed by models, and I agree with this. However, it is not direct to claim that a linear filter over an external heterogeneity is all that it takes to reproduce them. The dynamics of such a macroscopic system (vegetation cover) are highly nonlinear and hardly mimicked by a linear process. Furthermore, it is known that linear processes either diverge or collapse to zero, making them unsuitable to describe the formation of patterns.

Additionally, it is not clear to me whether the external heterogeneity is fixed on time or time-dependent, as the authors claim that they impose a mask on the nonlinear models, but also mention that constant perturbations will balance with filtering, finally reaching an equilibrium. Again, I disagree with this, as a linear process does not reach equilibria as mentioned above.

The authors mention that in an isotropic environment, the phase in the Fourier transform has no effect. I think this is not true, as the phase dictates what pattern one will observe. Take for example a labyrinth and a disordered hexagonal pattern of the same wavelength, they will show a similar power spectrum, but the phase is what determines their morphology (labyrinthine patterns have a much more disordered phase).

I found it weird that the authors declare contributions for four authors, but only three are listed on the manuscript. The same happens on the supplementary material, which shows four authors, in discrepancy with the authors listed in the manuscript. This supplementary material is also hard to read, with incomplete phrases that abruptly end, such as the one at the end of section 2 of the supp. Material.

Overall, I feel that the methods are not clearly explained, and the conclusions, such as “…we further demonstrate that patterns form through filtering of the noise…” require more study to support them, as again, those systems are highly nonlinear. This is readily seen by the authors when they themselves increase the ‘noise intensity’, seeing worse matches between the nonlinear models and the linear filters.

For these reasons, I can not recommend the article for publication.

Reviewer #2: Report on PONE-D-24-45670

Title: “Spatial pattern formation in heterogeneous environments”

The authors investigated a novel understanding of spatial pattern formation in ecosystems by incorporating

stochastic processes and scale-dependent feedbacks and highlighted how these patterns can be influenced by

various factors, such as noise and scale-dependent biophysical feedbacks. However, there are some issues that

need to be addressed carefully. The comments can be found below. I recommend revisions.

1 Comments

Comment 1. Natural ecosystems are rarely isolated. External influences, boundary effects, and environmental

gradients (e.g., climate or topography) may significantly influence pattern formation but appear underemphasized in the framework. In addition, temporal dynamics are also crucial in understanding ecosystem functioning. The role of temporal changes in pattern formation is not explicitly addressed in the article. It is

recommended that these issues briefly discussed as potential directions for future works.

Comment 2. The whole text needs to be checked in terms of typos and punctuation errors.

Reviewer #3: The study by Kästner et al. deals with the formation of spatial patterns in the biomass density of some species (e.g. vegetation in arid systems) in the presence of environmental heterogeneity. The authors claim that models that explain pattern formation through self-organization but ignore environmental heterogeneity predict idealized patterns that are far more regular than the patterns observed in nature and that by taking environmental heterogeneity into account, both regular and irregular patterns can be predicted that resemble their natural counterparts more closely. The most important result the authors present is that in heterogeneous environments the process of pattern formation can be understood as a filtering of the random spatial heterogeneities. In case of regular patterns, scale-dependent feedbacks act as a band-pass that amplify components of the noise (i.e., the random environmental heterogeneities) corresponding to the system's characteristic wavelength while suppressing all other components (on both shorter and longer spatial scales). In case of irregular spatial patterns, diffusion is shown to act as a low-pass that filters out short-scale components of the noise.

Overall, I consider this manuscript to be an interesting and important contribution to the literature on spatial pattern formation in ecological systems. It is very well written and meets, as far as I can tell, all publication criteria of PLoS ONE. I have a number of rather minor comments the authors may find helpful to further improve the presentation of their work.

1) The introduction is very accessible and provides a good overview of the relevant literature. To further improve it, the authors might provide an intuitive description of what "filtering" means in the context of their study. Since this is such a central concept in this study (and it certainly becomes clear later on), readers not familiar with it might benefit from some verbal explanations early on.

2) In the paragraph following line 81 it could be mentioned that the positive short-range feedback in vegetation models is due to improved water infiltration by established vegetation. In line 185 the authors mention infiltration without ever having stated its significance.

3) line 106: this looks like an incorrect figure reference. I guess the authors mean Fig. 1b_ii,c_ii. In general, panels of figures are not labeled consistently. Often, letters (a, b, c) with additional numerals (i, ii, iii) are used (e.g. Fig. 1), sometimes only letters (e.g. Fig. 3), and in Fig. 5 even a mix of both styles. At times, even the authors get confused by this (see e.g. line 383 or line 395). I recommend to use one style consistently.

4) Figure 1, caption: typos in the second line (a_i instead of a_ii) and at the very end (unnecessary 'et al.'). Also, I noticed that in this figure, 'autocorrelation' was used for the iii)-panels, whereas later (e.g. Fig. 3) 'correlogram' was used. Sure, the terms mean the same thing, but why not make it easier for the readers and use one term consistently?

5) line 211 and Fig. 2: it is never clearly stated how the 'mirror image' is defined.

6) lines 270-274: It looks like a few things got confused here. First, the sentence starting 'The characteristic wavenumber...' should probably start with 'In anisotropic patterns, the characteristic wavenumber...' to clearly separate it from the text preceding it (which is about isotropic patterns). The following sentence should be adjusted accordingly. Also, the authors first use 'x' for the direction perpendicular to the stripes (line 271/272) and immediately thereafter 'x' is the direction parallel to the stripes and 'y' is perpendicular to the stripes (line 273/274).

7) First paragraph of Section 3.1: While I agree with the authors that keeping most of the mathematics out of the main text improves it readability and accessibility, I still think that providing the main equations of the May and Rietkerk models (i.e., Equations 1 and 5da-5de in the Appendix) in the main text would be helpful. The authors frequently refer to parameters of these models, which is a bit awkward if the equations are not shown.

8) line 326: typo: 'are and are' -> 'and are'

9) Figure 5: the caption mentions a dashed horizontal line in panel a), but the line appears to be missing in the panel.

10) line 562: again, I recommend to use technical terms consistently throughout the manuscript. Here, 'spectral energy' is used for the first time, before the authors used 'spectral density'.

11) lines 619 and 649, Fig. 9 and 10: same thing, only here the authors use the term 'bands' to describe anisotropic patterns, everywhere else they use 'stripes'. Please unify!

12) Figure 10, panel f: I think the title should be 'Perpendicular to bands' (or rather: ... stripes) and the label on the y-axis should be 'Autocorrelation R_x'.

13) line 694: include a reference to Fig. 7b_i here, as there patterns generated with the Rietkerk model are shown. Also, the statement of the following sentence ('The spectral density ...') is not supported by Fig. 1c_iii alone. To which results should the readers compare this?

14) The numbering of the equations in the Appendix needs to be fixed.

6. PLOS authors have the option to publish the peer review history of their article (what does this mean?). If published, this will include your full peer review and any attached files.

Reviewer #1: No

Reviewer #2: No

Reviewer #3: No

---

## [Author Response · Author response to Decision Letter 1]

28 Jan 2025

Please see the attached PDF file with our detailed response. Note that the submission system put our response at the end of the merged PDF.

---

## [Decision Letter · Decision Letter 1]

13 Feb 2025

PONE-D-24-45670R1Spatial pattern formation in heterogeneous environmentsPLOS ONE

Dear Dr. Kästner,

Thank you for submitting your manuscript to PLOS ONE. After careful consideration, we feel that it has merit but does not fully meet PLOS ONE’s publication criteria as it currently stands. Therefore, we invite you to submit a revised version of the manuscript that addresses the points raised during the review process.

We look forward to receiving your revised manuscript.

Kind regards,

Pan Li, PhD

Academic Editor

PLOS ONE

Journal Requirements:

Reviewers' comments:

Reviewer's Responses to Questions

**Comments to the Author**

1. If the authors have adequately addressed your comments raised in a previous round of review and you feel that this manuscript is now acceptable for publication, you may indicate that here to bypass the “Comments to the Author” section, enter your conflict of interest statement in the “Confidential to Editor” section, and submit your "Accept" recommendation.

Reviewer #1: (No Response)

Reviewer #2: All comments have been addressed

2. Is the manuscript technically sound, and do the data support the conclusions?

Reviewer #1: Partly

Reviewer #2: Yes

3. Has the statistical analysis been performed appropriately and rigorously? 

Reviewer #1: I Don't Know

Reviewer #2: Yes

4. Have the authors made all data underlying the findings in their manuscript fully available?

Reviewer #1: Yes

Reviewer #2: Yes

5. Is the manuscript presented in an intelligible fashion and written in standard English?

Reviewer #1: Yes

Reviewer #2: Yes

6. Review Comments to the Author

Reviewer #1: I thank the authors for the revised version. The manuscript is generally in a much better shape, being readable and understandable. Now that I better understand the objective of the authors, I must highlight the following issues I have with the manuscript:

--. I think there is a confusion throughout the manuscript when referring to the effect of the heterogeneity as stochastic. While it is true that the authors employ heterogeneity ‘masks’ generated by a random process (this is what I mean by mask, the function describing the heterogeneous parameter), this does not render the dynamic stochastic. The dynamical system remains deterministic, as there is no fluctuating in time forcing. In other words, the system is not a stochastic differential equation, it remains a partial differential equation (or ordinary differential equation when space is discretized). This should be made clear, as various other works study the effect of stochasticity in its own, which is a problem tackled with different techniques.

--. The explanation of the linearization of the system is still not satisfactory to me. I must highlight that the eigenvalues of the Jacobian ( \partial A / \partial z ) do not need to be negative if you are linearizing around an arbitrary state. In particular, the case of pattern-forming systems makes this approach ambiguous. The authors state that they linearize around a homogeneous state. In a pattern forming system in the pattern forming regime, this will lead to eigenvalues both negative (for long and short wavelength perturbations) and positive (for perturbations around the Turing wavelength). Then, employing initial conditions with a non-negative projection in the unstable directions of the phase space will make the linear system diverge (even in the presence of spatial heterogeneities). The authors must acknowledge this limitation, and should be aware that their approach is correct only for linearizations very close to the equilibria of the nonlinear system (in the example I elaborated, near a patterned state and not a homogeneous state).

--. My previous point highlights why I stressed so much the importance of nonlinearities. Linearizing around a homogeneous state in a pattern-forming system will render equation (4) confusing. It indeed predicts an equilibrium, but it may be an unstable one. Thus, the authors might be reporting unstable states throughout the manuscript. The authors must take special care when explaining this.

This problem is avoided when applying the method to a non-pattern forming system. There, linearizing around a homogeneous state ensures the whole spectrum of eigenvalues is negative. There is a study where researchers analyze the effect of spatial heterogeneities in the parameters on a non-pattern forming model and find that heterogeneities explain many of the features exhibited in real vegetation covers, such as the power spectrum, the correlations, and even the patch size distribution [*]

[*]: Chaos, Solitons & Fractals 163, 112518 (2022)

--. I remain curious about the authors’ choice to neglect the phase of the Fourier transform of patterns. My main issue is regarding labyrinthine patterns, which are ubiquitous in vegetation cover patterns. They also fall inside the category of isotropic patterns, but are much more complex than hexagonal ones. The authors must discuss how their method applies to labyrinthine patterns. Maybe the authors will find relevant a study where the role of spatial heterogeneities in the parameters is studied in the labyrinthine patter-forming regime of vegetation cover models [**]. There, it is shown how changing the intensity of heterogeneities and their correlation length (they have a spatial structure) affects the power spectrum of such patterns, showing that they can be ideal, irregular, or scale-free. This highlights that heterogeneous parameters in nonlinear models are highly relevant to understand vegetation cover patterns.

[**]: Physical Review E 107 (5), 054219 (2023)

--. In line with my points regarding the importance of nonlinearities in the study of vegetation patterns, recent studies have shown how including heterogeneous parameters in models of vegetation cover modifies the diagram of states, that is, the vegetation pattern morphology as a function of driving parameters such as the aridity. There, researchers show that irregular and regular patterns could be alternative stable states induced by heterogeneity [***], highlighting the relevance of nonlinearities. Then, authors must be careful when presenting their work, as filtering could be a useful idea to understand the phenomenon, but only (very) close to already known equilibrium states.

[***]: arXiv preprint arXiv:2406.12581 (2024)

Overall, I think the manuscript has merit and could make a relevant piece of work for the community. However, the authors must acknowledge the weaknesses of their approach more carefully and not deem the nonlinear nature of these complex systems as a burden, as it can be highly relevant even when the current models used to study it are just qualitative or phenomenological. For these reasons, I can not recommend to publish this manuscript.

Reviewer #2: (No Response)

7. PLOS authors have the option to publish the peer review history of their article (what does this mean?). If published, this will include your full peer review and any attached files.

Reviewer #1: No

Reviewer #2: No

---

## [Author Response · Author response to Decision Letter 2]

31 Mar 2025

Please see the attached PDF for our detailed response. The response starts on page 122 of the merged document.

---

## [Decision Letter · Decision Letter 2]

22 Apr 2025

Formation of spatial vegetation patterns in heterogeneous environments

PONE-D-24-45670R2

Dear Dr. Kästner,

We’re pleased to inform you that your manuscript has been judged scientifically suitable for publication and will be formally accepted for publication once it meets all outstanding technical requirements.

Kind regards,

Pan Li, PhD

Academic Editor

PLOS ONE

Additional Editor Comments (optional):

Reviewers' comments:

Reviewer's Responses to Questions

**Comments to the Author**

1. If the authors have adequately addressed your comments raised in a previous round of review and you feel that this manuscript is now acceptable for publication, you may indicate that here to bypass the “Comments to the Author” section, enter your conflict of interest statement in the “Confidential to Editor” section, and submit your "Accept" recommendation.

Reviewer #1: (No Response)

2. Is the manuscript technically sound, and do the data support the conclusions?

Reviewer #1: Yes

3. Has the statistical analysis been performed appropriately and rigorously? 

Reviewer #1: I Don't Know

4. Have the authors made all data underlying the findings in their manuscript fully available?

Reviewer #1: Yes

5. Is the manuscript presented in an intelligible fashion and written in standard English?

Reviewer #1: Yes

6. Review Comments to the Author

Reviewer #1: I thank the authors for the re-revised version of the manuscript.

As a small clarifying comment, while it is true that the autocorrelation and the spectral density are Fourier transform pairs and that a power-law spectral density is a power-law autocorrelation, it is still possible to have an exponential autocorrelation and a spectral density with power-law behavior for the high-frequency components. This is because exponential autocorrelation gives a Lorenzian-like shape, which, for short wavelengths, has power-law scaling.

With that being said, I think the authors have addressed all my comments. The manuscript is now much more accessible, and in my opinion, it is a nice contribution towards understanding vegetation patterns. I am happy with the publication of the manuscript in PLOS One.

7. PLOS authors have the option to publish the peer review history of their article (what does this mean?). If published, this will include your full peer review and any attached files.

Reviewer #1: No

---

## [Editor Report · Acceptance letter]

PONE-D-24-45670R2

PLOS ONE

Dear Dr. Kästner,

I'm pleased to inform you that your manuscript has been deemed suitable for publication in PLOS ONE. Congratulations! Your manuscript is now being handed over to our production team.

Kind regards,

on behalf of

Dr. Pan Li

Academic Editor

PLOS ONE